# A spatial multicriteria prioritizing approach for geo-hydrological risk mitigation planning in small and densely urbanized Mediterranean basins

Guido Paliaga[1], Francesco Faccini[2], Fabio Luino[1], Laura Turconi[1]

[1]CNR IRPI Research Institute for Geo-Hydrological Protection – Strada delle Cacce 73, 10135 Torino (Italy)
[2]UNIVERSITA' DI GENOVA – DISTAV Department of Earth, Environmental and Life Sciences, Genoa University (Italy)

*Correspondence to*: Fabio Luino (fabio.luino@irpi.cnr.it)

**Abstract**

Landslides and floods, particularly flash floods, occurred currently in many Mediterranean catchments as a consequence of heavy rainfall events, causing damage and sometimes casualties. The high hazard is often associated with high vulnerability deriving from an intense urbanization in particular along the coastline where streams are habitually culverted. The necessary risk mitigation strategies should be applied at catchment scale with a holistic approach, avoiding spot interventions.

In the present work a high-risk area, hit in the past by several floods and concurrent superficial landslides due to extremely localized and intense rain events, has been studied. 21 small catchments have been identified: only some of them have been hit by extremely damaging past events, but all lies in the intense rain high hazard area and are strongly urbanized in the lower coastal zone. The question is what would happen if an intense rain event should stroke one of the not previously hit catchment; some situations

could be worse or not, so the attention has been focused on the comparison between catchments. The aim of the research has been identifying a priority scale between catchments, pointing out the more critical ones and giving a quantitative comparison tool for decision makers to support a strong scheduling of long-time planning interventions at catchment scale. The past events effects and the geomorphic processes analysis together with the field survey allowed to select three sets of parameters: one describing the

morphometric-morphological features related to flood and landslide hazard, another describing the degree of urbanization and of anthropogenic modifications at catchment scale and the last related to the elements

that are exposed to risk. The realized geodatabase allowed to apply the spatial multicriteria analysis technique (S-MCA) to the descriptive parameters and to get to a priority scale between the analyzed catchments. The scale can be used to plan risk mitigation interventions starting from the more critical catchments, then focusing economic resources primarily on them and obtaining an effective prevention strategy. The methodology could be useful even to check how the priority scale is modified during the progress of the mitigation works realization.

Besides, this approach could be applied in similar context, even between sub-catchments, after identifying a suitable set of descriptive parameters depending on the active geomorphological processes and the kind of anthropogenic modification. The prioritization would allow to invest economic resources in risk mitigation interventions priory in the more critical catchments.

## 1 Introduction

Floods and landslides are very common in many areas of the Mediterranean basin inducing a high geo-hydrological hazard (Canuti et al., 2001; Guzzetti and Tonelli, 2004; Luino, 2005; Luino and Turconi, 2017) and causing many casualties and significant damages every year. The 2017 periodic CNR-IRPI report (CNR, 2018; Brunetti et al, 2015) on Italian population landslides and floods threat evidences 1,789 casualties and 317,526 homeless in the period 1967-2016, with all the regions affected. Liguria, despite its small surface, is between the most affected region scoring the third place in the mortality index calculated on both landslide and flood events.

Among the geo-hydrologic processes, flash floods are the most hazardous for the short development time that often do not allow the population to protect itself. They occur following very intense and localized rainfall events and their ground effects have been underlined by many authors (Roth et al., 1996; Massacand et al., 1998; Delrieu et al., 2006; Amengual et al, 2007; Gaume et al., 2009; Marchi et al., 2009; Barthlott and Kirshbaum, 2013; Faccini et al, 2018). Spread shallow landslides and debris/mud flow often occur and their effects are superimposed and may locally magnify flooding, in particular in urban/suburban areas (Borga et al., 2014). Small catchments have a quick response to those events, reacting with large discharge of water and debris to the usually densely urbanized floodplain (Pasche et

al., 2008; Gaume et al., 2009). Many coastal Mediterranean areas are particularly liable to this kind of hazard: the general climatic context, with the interface between cold air masses and the sea, a steep territory and a complex geologic and geomorphologic context are the main natural factors. In such hazardous context the high vulnerability that characterizes most of the urbanization determines the elevate risk, while the intense anthropogenic modification of large portion of catchments and of hydrographical networks tends to amplify the effects (Tropeano and Turconi, 2003; Nirupama et al., 2006; Audisio and Turconi, 2011; Petrea et al, 2011; Llasat et al., 2014; Faccini et al, 2018; Acquaotta et al, 2018b): impervious surfaces, induced by soil consumption and urban sprawl, increase the surface run-off and decrease the time of concentration (Shuster et al., 2007), while strictly constrained and often culverted riverbeds have frequently inadequate discharge capacity (Moramarco et al., 2005; Faccini et al, 2015; Faccini et al., 2016).

Furthermore, the modifications are often interesting even the hinterland: besides urban sprawl and fragmentation caused by infrastructures, in some areas the ancient man-made terraces realized for agricultural practice and actually largely abandoned, constitute an increasing factor of geomorphological hazard (Brancucci and Paliaga, 2006; Tarolli et al., 2014; Paliaga, 2016). In recent years many evidences have been arising in Italy: large areas of Liguria (Brandolini et al., 2017; Cevasco et al., 2017) and Toscana (Bazzoffi and Gardin, 2011) are interested by terraces instability that may turn into source of geomorphologic hazard. In the Mediterranean region many areas present similar occurrence of terraces with analogous problems: the French Côte d'Azur, the Mediterranean and insular Spain and Greece (Tarolli et al., 2014) are some example. In the recent years some disastrous events involved terraced slopes: in 2011, during the Cinque Terre flood (Liguria, northern Italy) (Brandolini et al., 2017; Luino and Turconi, 2017), many terraces collapsed and the subsequent debris filled villages at a height of about 3 m, and in 2014, in the Leivi village during the Chiavari flood (Liguria) a terraced slope collapsed destroying a house and causing 2 fatalities (Faccini et al., 2017; Luino and Turconi, 2017).

Within this framework risk mitigation strategies are more and more urgent but largely disregarded, unapplied or only partially pursued: few resources are allocated and, commonly, are used only for emergency actions while a long-term planning and scheduling should be crucial to obtain significant results (Prenger-Berninghoff et al., 2014). In the recent years, in Italy, some large structural works have

been started to mitigate the worst flooding risk situations, but without following a broad approach at catchment scale. The most important is the floodway channel for the Bisagno stream in Genoa (Liguria), but similar project or culvert adjusting are ongoing in smaller neighboring streams. This approach allows
to reduce just a part of the risk, ignoring slope instability processes and related contribution to solid transport into hydrographical network.

Liguria, and especially the Genoa metropolitan area, are paradigmatic of the mixing of high hazard, with heavy rainfall that appear to be increasing in intensity (Faccini et al., 2015; Aquaotta et al., 2018a), elevate exposure at risk and lack of long-time planning mitigation strategies at catchment scale.

Apart the structural interventions in the larger Bisagno catchment, even the smaller ones in the Genoa metropolitan area are considered at high risk from the local environmental agency (ARPAL, Agenzia Regionale per la Protezione dell'Ambiente Ligure – Ligurian Environment Protection Agency) and would request mitigation works to be planned and scheduled.

The aim of the research is to propose a quantitative support tool to decision makers in order to plan and
schedule long-term interventions, identifying a priority scale between small catchments: their number and the different features that characterizes them request a comparison tool in order to evaluate the ones that are more critical. A group of 21 small catchments in the middle of the zone more liable to heavy rainfall (Cassola et al., 2016) have been analyzed, comparing three sets of descriptive parameters. The comparison has been performed with spatial multicriteria analysis (S-MCA) using a total of 19 parameters and
obtaining a priority scale between the 21 catchments. Spatial multicriteria approach has been applied by many authors in flood risk and in natural hazard management (Gamper et al., 2006; de Brito et al., 2006), mostly to identify flood prone areas and flood risk assessment (Fernández and Lutz, 2010; Wang et al, 2011), landslides susceptibility assessment (Feizizadeh and Blaschke, 2013; Nsengiyumva et al, 2018) or to compare catchments through morphometric parameters (Benzougagh et al., 2017). S-MCA techniques
are widely applied as decision support system in planning and environmental sustainability decision making to compare different design choices or site selection (Jacek, 2006; Bagli, 2011).   In the present work the Authors applied S-MCA techniques considering a broad set of parameters and trying to address the peculiarity of highly modified small urban catchments in a mountainous territory where comparing different sets of parameters describing different and inhomogeneous features appears crucial. The rank

obtained with the methodology could be used to evaluate the catchments that need more urgent actions in order to mitigate future eventual damage and casualties, considering that past extreme rainfall events hit bordering ones but, in the future, could replicate their effects. Then the necessary long-time planning could focus economic resources mainly on the more critical catchments, while the analysis of the descriptive parameters would be a support for pointing out the specific criticalities and then to design the

interventions.

## 2 Material and method

### 2.1 Geomorphological and geological settings

The studied area is one of the most critical in terms of geo-hydrological risk in Italy and in the Mediterranean basin, due to the morphometric features and to the high urbanization. It is located in the

central part of Liguria region, northern Italy (fig.1): 21 catchments with a surface area comprised between 1.3 and 27.5 km$^2$ have been analyzed. Five of them, numbered 11, 13, 14 and 15 in fig. 1, are sub-catchments of the two major ones that cross Genoa city: the Bisagno and Polcevera catchments. The confluence of n° 13 with Polcevera is just upward the already collapsed Morandi bridge. All the others flow directly into the Ligurian sea.

The area is densely populated, 2,429 inhab/km$^2$ in the whole Genoa administration unit (ISTAT, 2012) and has been strongly urbanized starting from the beginning of the 20$^{th}$ century (Faccini et al., 2016). Land use (fig. 1) clearly shows the strong dualism between the urban area, mainly concentrated in the lower catchments close to the sea, and the middle and upper mountainous catchments that preserve natural features with meadows and woods. Some catchments have been strongly modified by urbanization: in

particular n° 8, 9, 10, 12, 15 and 16. In the upper part of catchments 11, 12 and 13 the natural features and the presence of cultural heritages is testified by a highly frequented urban park. (Sacchini et al, 2018). Neotectonics activity has deeply influenced the structural asset, catchments' morphometry and hydrographical network features (Paliaga, 2015). The catchments are mainly elongated and oriented orthogonally to the coast line and reach maximum altitudes comprised between 491 and 1189 m a.s.l.

(tab. 1). Only n° 1, 3 and 4 present a less elongated feature. The strong steepness of the slopes and a

substantial lack of coastal floodplain is a distinctive feature of all the area: slope gradient is high in all the catchments and particularly in n° 3 and 21 (fig. 2). The only relatively extended floodplains are present in catchments n° 8, 9, 10, 14 and 16.

The catchments present substantial homogeneous lithological features if considered in three groups (fig. 3): the western one (from n° 1 to 7) are prevalently ophiolitic and metamorphic; the eastern (from n° 11 to 21) are essentially sedimentary, while the central ones (from n° 8 to 10) present both lithologies. Hydrographical networks are generally well developed (tab. 1), but present a higher density in the western catchments, due to the more impervious substrate. Main streams are generally short, coherently with the small dimensions of the catchments. Almost all the final stretches of the main streams have been culverted due to the dense urbanization: the only exceptions are n° 3, 11 and 19. In fig. 1 culvert in the final 1 km stretches are shown. Data of the floods that hit the catchments in the period 1950-2016 (Guzzetti et al., 1994; Luino and Turconi, 2017) are reported in fig. 4 and demonstrate the high geo-hydrological risk in the area. Some recent events resulted particularly dramatic: 1 casualty in n° 10 in 2010 and 6 casualties in n° 15 in 2011.

Landslides are widespread along most of the catchments (fig. 5); most of the processes are shallow and, despite the small dimension, sometimes they may produce high local damage, interacting with infrastructures and urban area. In occasion of flash floods that hit the area (i.e. in 2010, 2011, 2014 and 2015) high solid transport, supplied by superficial landslides, occluded partially or totally some culverts, contributing significantly to the streams overflow. In the area are present even some large DSGSD (Deep Seated Gravitational Slope Deformation) and an ancient landslide dam in n° 14.

Anthropogenic modification has interested even the not urbanized area: in the past, due to the high gradient and to the need of subsistence agricultural practices, slopes were widely modified by man-made terraces (fig. 6). The structures are largely abandoned and affected by instability and erosion, increasing the geo-hydrological hazard (Brancucci and Paliaga, 2006; Tarolli et al., 2014; Paliaga, 2016). Recent events in the Cinque Terre (2011) and in Leivi (Genoa metropolitan area, 2014) show the dramatic effects related to the presence of terraces and of their partial or total abandon (Cevasco et al., 2017; Giordan et al. 2017): widespread damage in the first, and two casualties in the latter.

## 2.2 Climate and Meteorological context

Climate is humid-mild with a short dry summer season (Sacchini et al., 2012; Acquaotta et al., 2018a), with annual mean rainfall between 1,100 and 1,300 mm and 14-16 °C annual mean temperature, registered in the 1945-2015 period. The impact of intense extreme events characterizes the area, mostly due to the cyclogenesis over the Ligurian Sea (Saéz de Càmara et al., 2011). This phenomenon is enhanced by the interaction between the general air mass circulation and the orography, characterized by high gradient slopes and the short distance of the mountains from the sea: the severe thermodynamic contrast between hot humid Mediterranean and colder continental air masses generates this configuration in the autumn-winter and spring periods (Anagnostopoulou et al., 2006), when thunderstorm convective systems and sometimes super-cells are triggered (Silvestro et al., 2012, 2016). Perturbations are canalized through the valley, causing very localized phenomena. During recent heavy rainfall events the maximum intensity registered was 180 mm/h in 2011 (Acquaotta et al., 2018b) and 140 mm/h (Faccini et al. 2016), respectively close and into catchment n° 15. During the 1970 flood event that hit Genoa area causing damage and 44 casualties, intensities over 200 mm/6h and over 500 mm/24h were registered (Faccini et al. 2016).

## 2.3 Research methodology

In order to support the decision process in planning reduction strategies of geo-hydrological risk, a comparison tool has been developed. The problem of relating heterogeneous physical quantities has been faced using the spatial multicriteria analysis techniques (S-MCA), commonly used as a support in decision making procedures, but applied even in natural hazard management (Gamper et al., 2006). The basic idea is to use a tool developed to compare heterogeneous physical quantities in order to obtain a sustainability scale between different alternatives to perform a priority scale of attention for the small catchments in term of geo-hydrological risk. The methodology considers parameters as gain or cost, depending on the influence they have in terms of sustainability: in the present study gain is intended as increasing hazard, while cost to lowering it. The selected parameters, due to their respective nature, have been considered as gain except for the concentration time, as its higher value determines a lower hazard

factor. Then the obtained rank puts at the higher level the catchments that have the higher gain, that is the

ones to be considered more critical from comparing all the selected parameters.

Considering the peculiarity of the studied area three sets of describing parameters at catchment scale have been selected: the first related to the natural features connected to geo-hydrological conditions, the second to the anthropogenic modification connected to hazard and the third to the exposure to risk, according to the flood directive 2007/60/EC.

The parameters selection has been performed considering both previous studies (Cevasco et al., 2017; Giordan et al. 2017; Faccini et al, 2018) and the active geomorphic processes in the catchments as they arise from the direct field survey dedicated mainly to point out instability processes active on the slopes and the possible sources of shallow landslides, the effects of intense rain events phenomena occurred in the recent past (2011, 2014, and 2015 events) and the diffuse inadequate size of culverts in the riverbeds.

Morphometric parameters defining the potential susceptibility of generating debris/mud flow and the ones related to flood potential have been selected from the related bibliography according to the field survey.

The level of anthropogenic modification has been defined through parameters that involve surface imperviousness, riverbed culvert and the presence of terraces, which are prevalently abandoned; in particular the culverting of the final stretch of the riverbeds often shows inadequacy in case of heavy rains

when the water flow, solid and floating transport reach their maximum transport capacity.

Exposure to risk is defined considering the elements that may be threatened by floods as they have been adopted by the local authority- Regione Liguria - after the hydraulic modeling, that is the hazard assessment, and the evaluation of the potential damage, then vulnerability. The official data define areas and punctual elements exposed to 4 increasing risk levels from R1 to R4.

The flow chart of the prioritizing process is shown in fig. 7 and the selected parameters are as follows:

- Set 1 (environmental factors-natural evolution – tab.2):
  - Drainage density: it is related to the flood potential (Patton and Baker, 1976).
  - Mean slope: it is related to the time of concentration in the catchment.
  - Melton ratio: it has been used as a potential indicator of susceptibility to generate debris
240         flow (Aversa et al., 2016).

- o Ruggedness number: it is related to flash flood potential and high erosion rate (Patton and Baker, 1976).
- o Hypsometric integral: it is correlated to the stage of geomorphic development of the catchment, is an indicator of the erosional stage and is related to several geometric and hydrological properties such as flood plain area and potential surface storage (Rogelis and Werner, 2014).
- o Landslides: total surface in percentage considering the catchment surface, excluding DSGSD.
- o Mean bifurcation ratio, obtained as the average value of the Rb for all stream orders: high values are correlated to flash flooding potential (Howard, 1990; Rakesh et al., 2000).
- o Times of concentration: the calculation has been performed with Pasini, Ventura, Pezzoli, Kirpich and NRCS-SCS formulae (tab.3); the mean value has been chosen. For NRCS-SCS application a prior CN evaluation has been assessed through land use data.
- o Flood hazard zone (200 years return period estimation) as the surface in percentage respect to the total catchment surface.
- Set 2 (environmental factors-anthropogenic impact):
  - o Soil consumption in percentage of the total catchment surface
  - o Culvert: percentage of the last km of the main stream.
  - o Terraces total surface in percentage respect to the catchment surface.
- Set 3 (elements to risk):
  - o Percentage of the area exposed to risk level R1.
  - o Percentage of the area exposed to risk level R2.
  - o Percentage of the area exposed to risk level R3.
  - o Percentage of the area exposed to risk level R4.
  - o Number of punctual elements exposed to risk level R2.
  - o Number of punctual elements exposed to risk level R4.

Considering the percentage of the catchment surface for the flood hazard zone (set 1) and for the area exposed to risk level R1-R4 (set 3) is similar to weighting with the catchment extension. Surface area, then, is implicitly part of the process of computation.

No punctual elements in the classes R1 and R3 are present in the studied catchments.

The descriptive parameters have been collected in a geodatabase related to catchments geometry in order to allow the application of S-MCA, performed through the Geo-UmbriaSUIT plugin (Massei et al, 2016) available in Quantum GIS free and open source software. The software performs a TOPSIS (Technique for Order of Preference by Similarity to Ideal Solution) multicriteria process (Triantaphyllou, 2000; Opricovic and Gwo-Hshiung, 2004); the method has been chosen among several ones for the good integration with the GIS environment. It has been originally elaborated to perform the ranking of different alternatives described by factors, aiming to the better one. In this study it has been applied to point out the catchments with the worst condition in terms of the selected parameters. Conceptually the application of the method does not change, even if the classification is done with the worst element at the top: a set of factors describing heterogeneous features is used to compare the described elements, that are the catchments. Then factors, defined as gain or cost depending on the positive or negative effect they have, and choices in the TOPSIS model become respectively parameters and catchments. The application is made considering factors that determine the worst conditions in terms of criticality of the catchments and the opposite significance between better and worst is only related to the values of the parameters: if they are related to an improving (gain) or worsening (cost) condition. Higher values in the chosen parameters, apart the time of concentration value, implies a worsening situation, then the ranking will classify at the first level the catchments in the worst situation.

To perform the computation of the parameters for the catchments in the study area the following vector and raster data, realized by Regione Liguria that is the regional authority, have been used:

- 5 m DTM (Digital Terrain Model) realized in 2007.
- Land use in scale 1:10000, realized in 2015.
- Landslides inventory from IFFI project (Inventario dei Fenomeni Franosi in Italia - Italian landslides inventory), updated in 2017, scale 1:10000.

•   Hydrographical network and culvert data from CTR (Carta Tecnica Regionale, Technical Regional Map) 1:5000, 2007.

     •   Flood data from the AVI (Aree Vulnerate Italiane da frane ed inondazioni – Floods and Landslides Damaged Italian Areas) archive (Guzzetti et al., 1994) for the period 1918-1998 and from the database of recent events in the period 2005-2016 (Luino and Turconi, 2017).

•   Aerial photography, shoot in 2014.

During the field survey of the whole area the ongoing risk reduction works that actually regards catchments n° 9, 10 and 16 with the stabilization of landslides, and n° 10 and 15 with structural works to the riverbed final stretch, respectively with the improvement of the culvert capacity and the realization of

an overflow channel, have been evaluated.

## 3 Results

The geodatabase, collected through the calculation of the 19 parameters and shown in table 4 and 5, evidences a certain variability of values. In table 6 the time of concentration values obtained with the different formulae are shown; for the S-MCA calculation the mean value has been chosen.

The results of the parameters computation give a descriptive scheme of the small catchments; some have similar characteristics, and some have specific peculiarities. All the catchments share high slope and hypsometric index values. Time of concentration is always short while landslide surface (%) shows a large variability as the value of the Melton ratio and drainage density.

Flood events interested 15 on 21 catchments and some of them have been repeatedly hit. Flood hazard

zones are quite extended in some cases and always involve densely populated areas.

Regarding catchments anthropogenic modifications, soil consumption is variable but always concentrated in the lowest part where are present even important infrastructures running along the coastline; in some cases, the value is particularly high. The highest quota slopes are usually in semi-natural conditions and in some catchments, man-made terraces are widespread and mostly abandoned. The final km culverted

percentage for the main stream assumes often high values, in some cases 100%. This modification represents one of the most critical as transport capacity is always inadequate in case of intense rain events,

causing flooding in the surrounding urban area. Besides buildings have been built close or, more frequently, over the cover.

The parameters describing the elements exposed to risk give an idea of the impact that a flood event may have on the urban area: both the percentage of the risk area, mainly residential, industrial and hospital, and the number of punctual elements, including schools and cultural heritages, are variously present but reach the highest values in catchment n° 9.

The analysis of data in the geodatabase evidences how catchment n° 9 often emerges for critical values, followed by n° 6, 8 and 17. Particular attention must be paid even on n° 11, a Polcevera's sub-catchment, and on 13, 14 and that are Bisagno's sub-catchments: in all these cases downward of the confluence with the main stream the urbanization degree is at the highest level with elevated population density and soil consumption. Recent flash flood events in 2011 and 2014 interested n° 13, 14 and 15 propagating the effects to Bisagno catchment. Other peculiarities are present in the n° 12: the largest of the small catchments that constitute the ancient Genoa amphitheater with the old harbor and the historical center. Finally, the western catchments show a lower soil consumption degree but larger widespread shallow areas of instability that during the recent intense rain events in 2011 and 2014 were activated.

But by leaving a qualitative approach for the quantitative one that is obtained by the application of the S-MCA techniques to compare the catchments' conditions, some more meaningful results may be obtained. The first application of the method has been performed without assuming different weights at *priori* to the describing parameters; even the same relative importance has been assumed for environmental factors (set 1 and 2) and for the elements to risk (set 3).The values obtained by the calculation have been ordered in 5 classes, being the number 1 the most critical, or the one that requests a higher level of attention for the risk reduction strategies. Results are shown in figure 8, while table 7 provides the score values obtained using all the parameters (priority scale A), only the anthropogenic origin ones (priority scale B) and only the natural origin ones (priority scale C) for the environmental factors. A further calculation has been performed assuming proportional weights to the elements to risk factors, that is giving a major importance to the higher risk level respect to the lower ones. The results are collected in fig. 9 and in table 7 and constitute the priority scale D.

## 4 Discussion

The results of the application of the S-MCA technique to the 21 small catchments represent an attempt to give a decision support tool to plan and manage investments for works aimed at mitigating geo-hydrological risk in an area hardly hit by floods, flash floods and landslides in the past, as addressed by many authors (De Brito et al., 2016). Ranking alternatives in flood and risk reduction strategies have been largely implemented and addressed to decision makers, using different S-MCA techniques (Andersson-Sköld et al, 2015; de Brito and Evers, 2016). The need for optimizing economic resources and to reduce risk is essential in critical situation with high inhabitants' density, strong anthropogenic modifications and characterized by a high hazard. Besides, flash flood events are strongly localized and in the recent years they hit prevalently some catchments (tab. 4): n° 4, 10 and 16 present the highest numbers, even if the most critical events happened in n° 8, 9, 10 and 15. Considering that all the studied area is characterized by high hazard for the possible hit of super-cell systems and presents high hazard even for the peculiar geomorphological features, the question is what would happen if a localized and intense event should hit every catchment. For this reason, and for the highly inadequate actual situation, it seems necessary to assess a priority scale considering both natural features of the catchments and the anthropogenic modifications that enhanced the risk level in order to obtain a priority scale on a quantitative base.

The priority scale A obtained evidences the critical situation of catchment n° 9 that emerged even at a qualitative analysis level, with n° 3 and 8 in the second rank and n° 1 in the third that were more difficult to recognize. These results suggest that, possibly, the highest attention in planning resources for risk reduction works at catchments scale should be paid to these higher-level ranks catchments. A detail study for the punctual activities would be essential, considering the activities at catchment scale.

Priority scale B and C have been obtained considering, respectively, only the anthropogenic parameters and only the natural ones, in order to evidence the different eventual influence of the two sets. Considering the scale C the natural tendency of catchments to geo-hydrological risk emerges a bit differently and, examining the scale A, a possible influence of anthropogenic modifications arises more clearly. Effectively catchments n° 8 and 9 have been particularly interested by human activities: the soil consumption is high, as high is the percentage of the final km culverted riverbed. We can deduce that

human interventions enhanced the most critical situations, while in other context the effect has been lower, even if always in the increasing direction.

The situation changes a little assuming a different weight to the elements to risk parameters, that is considering of proportional major importance the highest exposition to risk: the priority scale always sees catchments n° 3, 8 and 9 at the highest ranks, giving a further confirmation of how critical their situation is. At the opposite side of the priority scale, catchments n° 12, 18, 19, 20 and 21 are always stable in the lowest rank, meaning a possible lower level of attention, in respect to the other ones. For example, the

Fereggiano catchment (n° 15) critical situation is well known even at international level: the heavy rainfall in 2011 caused 6 casualties and much damage. Despite that it ranks at the 4th level in the priority scale. It does not mean that its risk level is not high, but that it has been hit by a heavy rainfall that caused a devastating consequence. If such an event would hit one of the other studied catchments, like n° 9 for example, the effect could be, probably, similar or even worse. At the same time the D scale shows that

catchments in the lower rank position are almost a half in respect to the ones at the same position in scale A.

Considering the high-risk level of the whole area the rank in the scale must be considered as an additional information: it does not mean that no reduction work should be performed in catchments at the lowest rank position, but only that the other ones should be considered more urgent.

Another consideration regards limitations in the approach related to peculiar situations that do not emerge from the comparison: in Geirato catchment (n° 14) is present a large landslide dam that is a potential source of high hazard, not limited to the catchment itself but possibly to the main Bisagno one. This limitation could be overcome by adding a parameter for punctual peculiar situations, but it has not been considered in the present work.

The prevention activity should include interventions on both streams and slopes, structural and non-structural: the inadequate transport capacity of culverted streams is always seen as the only problem to be solved but considering the high solid transport and debris/mud that often add their effect during the intense rain events and that act locally, interrupting roads or impacting buildings, and causing problems in the urbanized lower parts of the catchments, solutions should be studied holistically. The debate

between using structural or non-structural interventions for risk reduction has been faced by many authors

(Kundzewicz, 2002; Yazdi and Neyshabouri, 2012; Meyer et al, 2012) but in conditions like the studied one only the mutual concurrence of them may insure an acceptable result. A strong and continuous monitoring (Collins, 2008) and maintenance of the slopes, due to their straight closeness and relation with the urban area is crucial: from structural intervention on landslides stabilization to soil bioengineering techniques to reduce erosion and shallow landslides susceptibility and the recovery of abandoned terraces (Morgan and Rickson, 2003). The basic philosophy should be to act preventively on instability with even small and not invasive interventions widespread on the territory (Lateltin et al, 2005). These activities should be focused to reduce the potential debris and sediments that contribute substantially to saturate culverts during intense rain events. Considering that the critical situation deriving from the soil consumption cannot be modified, as re-naturalization is not an option considered acceptable both from decision makers and probably from large parts of the population, other interventions may be addressed to reduce the negative effects of the anthropogenic modifications. Only in very limited situations the eventual culvert elimination would be possible without knocking down buildings that is an option with a low acceptance level. In the other cases the possible solutions are structural hydraulic interventions that may guarantee the reduction of the extension of flood hazard zones and then even of the elements to risk areas. This include enlargement of embankments, restructuring of culverts and realization of diversion overflow channels. In the cases where these high cost interventions are crucial, like for catchments n° 8, 9, 10, 11, 12, 15 and 21, the reduction of solid transport in the streams, that is mainly reduction of erosion, shallow landslides ad stabilization of abandoned terraces, would contribute significantly to the risk mitigation. Cost of structural hydraulic interventions is usually high and of the order of millions of euro, while spread small interventions on the slopes are usually at less an order of magnitude lower, but the integration of the two is essential in many situations. For example, in catchments n° 9, 10, 14 and 15 where landslides, abandoned terraces and high gradient slopes are close and coupled with densely populated areas and intensely modified riverbed with inadequate capacity culverts. On the other hand, catchments n° 3, 4, 5 and 6 are mostly interested by slope instability processes and present a lower level of soil consumption and, more in general, of anthropogenic modifications.

The applicable mitigation measures present a good level of ecological compatibility, in particular the bioengineering ones along the slopes, for their low environmental impact, while the structural hydraulic

interventions would be done in urban areas producing only temporarily impacts on population, due to the construction site set-up. Regarding the potential acceptation of the population, the interventions along the slopes should not be problematic for their usually modest dimensions, while the structural hydraulic interventions higher impact, even if limited in time, and elevate cost could be a little more problematic. Actually, some important works are ongoing along the Bisagno stream, with traffic disturbance and influences on economic activities lasting for some years, but the population risk awareness has risen after the last devastating flash flood in 2011 and 2014.

Finally, risk reduction works would have a direct influence in the priority scale method: besides the stabilization of landslides, the structural interventions on streams would have the effect of modifying and reducing the extension of flood hazard zones and then even of the areas exposed at risk. In this way the methodology could be used even to simulate the effects of some structural important and expensive works on the overall rank in the priority scale. This information could be included in the cost/benefit analysis of the planned structural interventions.

**5 Conclusion**

Mitigation strategies for geo-hydrological risk request a catchment scale approach that results particularly crucial in a composite context where hazard related to natural features concur together with high anthropogenic modification of the territory and high vulnerability (Pasche et al., 2008). More in general prevention of geo-hydrological risk requests a decision-making process that is complex, affected by uncertainty (Akter and Simonovic, 2005; Kenyon, 2007) and often with limited economic resources at disposition.

Besides, an area characterized by many small urban catchments is complex to manage and a strong programming and planning is essential. The proposed method for prioritize planning for risk mitigation works between catchments could be used as a support tool to quantitatively address economic resources that usually are limited and request a strong optimization (Gamper et al., 2006). The approach could be even used in different context at sub-catchment scale to point out the more critical sub-catchment and basing the comparison on different sets of parameters depending on the active processes in the area. The procedure may be adapted and modified with weighting of selected parameters in order to give major

importance to the ones considered more important. Another adjustment of the method is possible considering the relative importance to the environmental set of parameters in respect to the elements to risk ones: depending on the value that we would assign to the different aspects of the evaluation, different weight may be assumed.

The application of the methodology in a high-risk area allowed to obtain a priority scale that is actually partially confirmed by the structural intervention that local authority is operating: some are in design phase and some are in construction. The critical situation of catchment n° 9 is actually being approached and the solution has been found in some important design for the adjustment of the culvert and of stream embankments; besides an overflow channel is going to be realized in the Bisagno catchment, involving

even the Fereggiano one (n° 15). These works are largely expensive but are now essential to reduce risk in a situation where the anthropogenic modification almost saturated all the available spaces in the floodplain, as it happened in all the small urban catchments examined in the present study. The risk reduction would require a holistic approach at catchment scale, considering all the processes acting, their mutual relationships and trying to address all the problems, considering that what happens along the

slopes influences even the lowest portion of the catchment itself (Samuels et al., 2006; Blöschl et al., 2013). Moreover, the cost of interventions along the slopes is usually significantly less economically impacting than the structural works are.

The cost of interventions has not been considered in the present study as the aim of the work was to compare the small catchments and realize a priority scale of attention to address planning on risk basis

but could be included in the methodology and perhaps developed in a subsequent phase. Its role would be at the same level of environmental and elements to risk factors and a weight could be assigned to find a balance among the three. Such evaluation could be done after a preliminary assessment of the interventions in all the comparing catchments; the application of the method in such a case could address more precisely the investment of economic resources.

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

**Figure 1**: Land use of the studied catchments (ref. to table 1). A: urban area; B: meadows; C: cultivations; D: woods; E: rocks and areas hit by fire.

**Figure 2**: Gradient in the studied catchments.

**Figure 3**: Simplified lithology of the studied catchments.

**Figure 4**: The hydrographical network with main streams culverted last stretch of the studied catchments; the light blue circles are proportional to the number of floods in the catchments in the period 1900-2016 (Guzzetti, 1994; Luino and Turconi, 2017).

**Figure 5**: Landslides in the studied catchments discriminated by activity status (IFFI database, 2017 update).

**Figure 6**: Man made terraces in the studied catchments.

**Figure 7**: The flow chart for the prioritizing method: the spatial multicriteria analysis allows to compare 3 sets of un-
homogeneous parameters to realize a classification of the catchments that can be used as a decision support system in risk mitigation planning.

**Figure 8**: The priority scale obtained using all the parameters, excluding DSGSD for the calculation of landslides.

**Figure 9**: The priority scale obtained using all the parameters, excluding DSGSD for the calculation of landslides and weighting the elements to risk factors.

**Table 1:** The main morphometric features of the studied catchments.

| Stream name | Catchment number | Area (km²) | Hydrographical network length (m) | Main stream length (m) | Mean altitude (m) | Minimum altitude (m) | Maximum altitude (m) |
|---|---|---|---|---|---|---|---|
| T. LERONE | 1 | 21.1 | 79150 | 8274 | 510 | 0 | 1189 |
| T. CANTARENA | 2 | 4.5 | 22573 | 4289 | 444 | 0 | 922 |
| T. CERUSA | 3 | 23.1 | 142921 | 7946 | 506 | 0 | 1177 |
| T. LEIRA | 4 | 27.5 | 144486 | 6249 | 410 | 0 | 1001 |
| T. BRANEGA | 5 | 4.7 | 26733 | 3339 | 290 | 0 | 859 |
| T. FOCE | 6 | 3.5 | 18629 | 3354 | 191 | 0 | 598 |
| T. VARENNA | 7 | 22.3 | 140566 | 10393 | 461 | 0 | 995 |
| R. MOLINASSI | 8 | 1.8 | 9246 | 3707 | 222 | 0 | 545 |
| R. CANTARENA | 9 | 1.9 | 5621 | 2443 | 131 | 0 | 435 |
| R. CHIARAVAGNA | 10 | 10.7 | 60531 | 6838 | 272 | 0 | 658 |
| T. TORBELLA | 11 | 5.0 | 21644 | 3946 | 232 | 14 | 635 |
| R. LAGACCIO | 12 | 3.4 | 7866 | 2773 | 199 | 0 | 493 |
| T. VELINO | 13 | 3.2 | 12439 | 3034 | 236 | 18 | 543 |
| T. GEIRATO | 14 | 7.8 | 27863 | 4368 | 296 | 47 | 779 |
| T. FEREGGIANO | 15 | 4.7 | 17197 | 4239 | 216 | 10 | 564 |
| T. STURLA | 16 | 13.3 | 54024 | 6995 | 316 | 0 | 845 |
| R. PRIARUGGIA | 17 | 1.5 | 3745 | 2680 | 145 | 0 | 491 |
| R. CASTAGNA | 18 | 1.4 | 5672 | 2652 | 165 | 0 | 540 |
| R. BAGNARA | 19 | 1.6 | 6816 | 2645 | 293 | 0 | 823 |
| R. S. PIETRO | 20 | 1.3 | 5940 | 2597 | 279 | 0 | 724 |
| T. NERVI | 21 | 9.0 | 51201 | 6166 | 391 | 0 | 846 |

**Table 2:** The morphometric parameters formulae used.

| Morphometric parameter | Formulae |
|---|---|
| Drainage density (km$^{-1}$) | $D_d = \dfrac{\sum L}{S}$ |
| Melton ratio | $Mi = (H_M - H_{Mm})/(S) \; \frac{1}{2}$ |
| Ruggedness number | $Rn = D_d * (H_M - H_m)$ |
| Hypsometric integral | $Hi = \dfrac{(H - H_m}{(H_M - H_m)}$ |
| Bifurcation ratio | $Rb = \dfrac{N_u}{N_{u+1}}$ |
| Catchment surface (km$^2$) | $S$ |
| Stream length (km) | $L$ |
| Strahler order | $u$ |
| Number of streams of order $u$ | $N_u$ |
| Main stream length (km) | $L_m$ |
| Main stream gradient (km/km) | $i$ |
| Mean elevation (km) | $H$ |
| Main stream difference in height (km) | $d$ |
| Maximum elevation (km) | $H_M$ |
| Minimum elevation (km) | $H_m$ |
| Medium elevation (km) | $H$ |
| Mean gradient of the slopes (%) | $y$ |


**Table 3:** Time of concentration formulae used.

| Time of concentration (h) | Formulae |
|---|---|
| Pasini | $t_c = 0.108 * \dfrac{(S * L_m)^{1/3}}{i^{1/2}}$ |
| Ventura | $t_c = 0.127 * (S/i)\ ½$ |
| Pezzoli | $t_c = 0.055 * \dfrac{L_m}{i^{1/2}}$ |
| Kirpich | $t_c = 0.095 * \dfrac{L_m^{1.155}}{d^{0.385}}$ |
| NRCS-SCS | $t_c = 0.57 * \dfrac{L_m^{0.8} * (X + 1)^{0.7}}{y^{1/2}}$ $X = \dfrac{1000}{CN} - 10$ $CN$= curve number |


**Table 4:** The geodatabase with the chosen criteria related to geo-hydrological hazard. The *a* through *j* parameters are related to natural features, while the *k* through *m* by anthropogenic modifications.

| Catchment number | a<br>Dd<br>(km⁻¹) | b<br>Mean gradient (%) | c<br>Mi | d<br>Rn | e<br>Hi | f<br>Landslide (%) | g<br>Rb mean | h<br>Time of concentration (') | i<br>Floods number | j<br>Flood hazard zone 200 y (%) | k<br>Soil consumption (%) | l<br>Culvert last km (%) | m<br>Terraces (%) |
|---|---|---|---|---|---|---|---|---|---|---|---|---|---|
| 1 | 3.75 | 56.5 | 0.26 | 4.45 | 0.43 | 0.2 | 0.28 | 82.64 | 2 | 0.3 | 10.9 | 5.1 | 12.1 |
| 2 | 4.96 | 55.9 | 0.43 | 4.58 | 0.48 | 0.4 | 0.25 | 36.15 | 0 | 0.6 | 17.1 | 25.0 | 14.1 |
| 3 | 6.19 | 60.4 | 0.25 | 7.29 | 0.43 | 6.6 | 0.31 | 81.34 | 1 | 0.7 | 7.4 | 7.4 | 19.2 |
| 4 | 5.25 | 62.1 | 0.19 | 5.26 | 0.41 | 4.3 | 0.32 | 73.37 | 20 | 0.2 | 20.7 | 10.5 | 20.1 |
| 5 | 5.71 | 46.1 | 0.40 | 4.90 | 0.34 | 6.5 | 0.24 | 29.53 | 3 | 0.6 | 27.8 | 9.0 | 9.9 |
| 6 | 5.34 | 45.4 | 0.32 | 3.19 | 0.32 | 4.9 | 0.26 | 35.29 | 4 | 0.5 | 9.5 | 22.2 | 40.3 |
| 7 | 6.30 | 56.0 | 0.21 | 6.27 | 0.46 | 0.6 | 0.30 | 110.08 | 6 | 0.3 | 16.9 | 11.4 | 9.6 |
| 8 | 5.06 | 47.2 | 0.40 | 2.76 | 0.41 | 0.0 | 0.21 | 33.80 | 2 | 3.4 | 20.4 | 45.9 | 18.8 |
| 9 | 3.01 | 31.8 | 0.32 | 1.31 | 0.30 | 0.0 | 0.07 | 27.16 | 4 | 10.6 | 49.4 | 34.4 | 6.6 |
| 10 | 5.65 | 49.4 | 0.20 | 3.72 | 0.41 | 0.1 | 0.29 | 77.65 | 17 | 2.7 | 23.4 | 17.6 | 5.3 |
| 11 | 4.33 | 46.3 | 0.28 | 2.69 | 0.35 | 0.8 | 0.29 | 39.58 | 1 | 1.9 | 13.6 | 0.0 | 18.0 |
| 12 | 2.33 | 45.1 | 0.27 | 1.15 | 0.40 | 0.0 | 0.31 | 34.43 | 0 | 0.1 | 36.3 | 100.0 | 0.0 |
| 13 | 3.84 | 55.0 | 0.29 | 2.02 | 0.42 | 2.8 | 0.31 | 35.75 | 1 | 1.8 | 7.3 | 35.7 | 5.6 |
| 14 | 3.58 | 49.9 | 0.26 | 2.62 | 0.34 | 0.2 | 0.37 | 50.12 | 2 | 0.6 | 7.7 | 11.8 | 29.2 |
| 15 | 3.68 | 48.2 | 0.26 | 2.04 | 0.37 | 0.0 | 0.30 | 55.11 | 4 | 3.4 | 19.0 | 80.4 | 26.8 |
| 16 | 4.05 | 50.6 | 0.23 | 3.42 | 0.37 | 0.0 | 0.32 | 85.44 | 10 | 2.0 | 13.8 | 9.8 | 16.7 |
| 17 | 2.58 | 32.6 | 0.41 | 1.27 | 0.30 | 0.0 | 0.13 | 26.56 | 0 | 0.5 | 34.0 | 17.2 | 32.3 |
| 18 | 4.04 | 38.6 | 0.46 | 2.18 | 0.31 | 0.0 | 0.39 | 26.06 | 0 | 0.0 | 22.3 | 3.6 | 32.3 |
| 19 | 4.35 | 50.8 | 0.66 | 3.58 | 0.36 | 1.3 | 0.22 | 19.56 | 0 | 0.1 | 15.1 | 10.7 | 14.6 |
| 20 | 4.47 | 55.3 | 0.63 | 3.23 | 0.39 | 0.0 | 0.32 | 20.38 | 0 | 0.0 | 8.6 | 18.2 | 12.8 |
| 21 | 5.66 | 65.8 | 0.28 | 4.79 | 0.46 | 0.7 | 0.29 | 65.20 | 7 | 0.4 | 3.3 | 100.0 | 11.5 |


**Table 5:** The geodatabase with the evaluation of the surfaces (%) and punctual elements to risk in the studied catchments, according to the EU Flood Directive 2007/60/CE.

| Catchment number | R1 risk area (%) | R2 risk area (%) | R3 risk area (%) | R4 risk area (%) | R2 risk elements | R4 risk elements |
|---|---|---|---|---|---|---|
| 1 | 0.19 | 0.02 | 0.00 | 0.16 | 0 | 0 |
| 2 | 0.14 | 0.61 | 0.02 | 0.48 | 1 | 0 |
| 3 | 0.16 | 0.18 | 0.04 | 0.53 | 6 | 0 |
| 4 | 0.08 | 0.18 | 0.00 | 0.14 | 2 | 1 |
| 5 | 0.04 | 0.41 | 0.01 | 0.38 | 1 | 1 |
| 6 | 0.07 | 1.03 | 0.00 | 0.45 | 0 | 0 |
| 7 | 0.13 | 0.37 | 0.00 | 0.17 | 1 | 0 |
| 8 | 0.20 | 2.30 | 0.00 | 3.21 | 0 | 2 |
| 9 | 0.02 | 0.24 | 0.00 | 10.54 | 0 | 13 |
| 10 | 0.07 | 0.57 | 0.04 | 2.61 | 0 | 3 |
| 11 | 0.08 | 0.70 | 0.08 | 1.73 | 0 | 0 |
| 12 | 0.00 | 0.00 | 0.00 | 0.14 | 0 | 0 |
| 13 | 0.01 | 0.60 | 0.73 | 1.05 | 0 | 0 |
| 14 | 0.08 | 0.97 | 0.02 | 0.52 | 1 | 1 |
| 15 | 0.00 | 0.28 | 0.05 | 3.30 | 0 | 2 |
| 16 | 0.27 | 0.64 | 0.06 | 1.70 | 0 | 0 |
| 17 | 0.11 | 0.15 | 0.02 | 0.50 | 0 | 0 |
| 18 | 0.00 | 0.00 | 0.00 | 0.00 | 0 | 0 |
| 19 | 0.00 | 0.05 | 0.00 | 0.11 | 0 | 0 |
| 20 | 0.00 | 0.01 | 0.00 | 0.02 | 0 | 0 |
| 21 | 0.02 | 0.09 | 0.01 | 0.35 | 0 | 0 |


**Table 6:** Time of concentration for the studied catchments: 5 methodologies have been used and the mean value has been chosen as representative in table 4.

| Catchment number | Pasini (m) | Ventura (m) | Pezzoli (m) | Kirpich (m) | NRCS-SCS (m) | Mean value (m) |
|---|---|---|---|---|---|---|
| 1 | 109.0 | 105.5 | 82.1 | 47.0 | 69.6 | 82.6 |
| 2 | 43.8 | 40.9 | 35.5 | 24.7 | 35.8 | 36.1 |
| 3 | 108.8 | 108.3 | 77.5 | 45.0 | 67.1 | 81.3 |
| 4 | 103.6 | 115.1 | 59.3 | 36.6 | 52.3 | 73.4 |
| 5 | 34.7 | 35.3 | 23.6 | 18.0 | 36.1 | 29.5 |
| 6 | 43.7 | 42.4 | 32.9 | 23.3 | 34.2 | 35.3 |
| 7 | 145.3 | 131.6 | 125.2 | 65.1 | 83.2 | 110.1 |
| 8 | 38.2 | 32.2 | 38.2 | 26.1 | 34.3 | 33.8 |
| 9 | 33.9 | 32.9 | 25.5 | 19.1 | 24.4 | 27.2 |
| 10 | 102.4 | 94.3 | 85.2 | 48.4 | 58.0 | 77.6 |
| 11 | 49.9 | 48.7 | 37.1 | 25.5 | 36.6 | 39.6 |
| 12 | 46.9 | 48.2 | 31.4 | 22.5 | 23.2 | 34.4 |
| 13 | 46.1 | 45.6 | 33.3 | 23.5 | 30.3 | 35.8 |
| 14 | 66.1 | 67.1 | 45.4 | 29.8 | 42.2 | 50.1 |
| 15 | 75.1 | 70.6 | 59.9 | 36.9 | 33.0 | 55.1 |
| 16 | 117.2 | 111.1 | 92.0 | 51.4 | 55.5 | 85.4 |
| 17 | 32.1 | 29.0 | 27.9 | 20.5 | 23.4 | 26.6 |
| 18 | 30.5 | 27.5 | 26.6 | 19.8 | 25.9 | 26.1 |
| 19 | 21.2 | 19.4 | 17.7 | 14.5 | 25.0 | 19.6 |
| 20 | 22.0 | 19.8 | 19.3 | 15.4 | 25.4 | 20.4 |
| 21 | 84.5 | 78.3 | 69.4 | 41.3 | 52.5 | 65.2 |


**Table 7:** The priority scales - A: using all the parameters; B: using parameters $k$, $l$ and $m$ (ref. Tab. 4); C: using parameters $a$ through $j$ (Tab. 2); D: using all the parameters and weighting the elements to risk ones (tab 3).

| Catchment number | Priority scale A | Priority scale B | Priority scale C | Priority scale D |
|---|---|---|---|---|
| 1 | 5 | 5 | 4 | 4 |
| 2 | 4 | 4 | 4 | 3 |
| 3 | 2 | 3 | 1 | 1 |
| 4 | 4 | 4 | 3 | 3 |
| 5 | 4 | 5 | 3 | 3 |
| 6 | 4 | 4 | 3 | 3 |
| 7 | 5 | 5 | 4 | 4 |
| 8 | 2 | 2 | 2 | 1 |
| 9 | 1 | 1 | 2 | 2 |
| 10 | 4 | 4 | 4 | 4 |
| 11 | 5 | 4 | 4 | 4 |
| 12 | 5 | 4 | 5 | 5 |
| 13 | 3 | 3 | 2 | 3 |
| 14 | 4 | 4 | 4 | 4 |
| 15 | 4 | 3 | 4 | 4 |
| 16 | 4 | 4 | 4 | 3 |
| 17 | 5 | 4 | 5 | 4 |
| 18 | 5 | 5 | 5 | 5 |
| 19 | 5 | 5 | 4 | 5 |
| 20 | 5 | 5 | 5 | 5 |
| 21 | 5 | 4 | 5 | 5 |

**Priority scale**

| |
|---|
| 1 |
| 2 |
| 3 |
| 4 |
| 5 |


# FIGURES

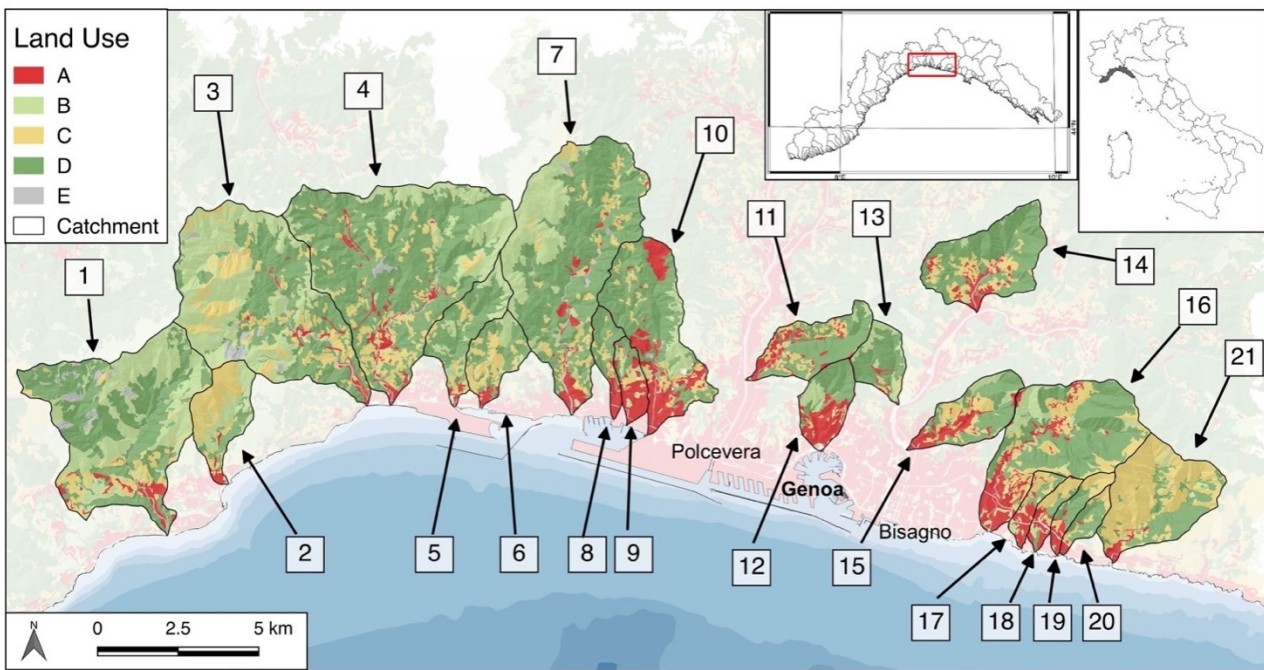

Figure 1:

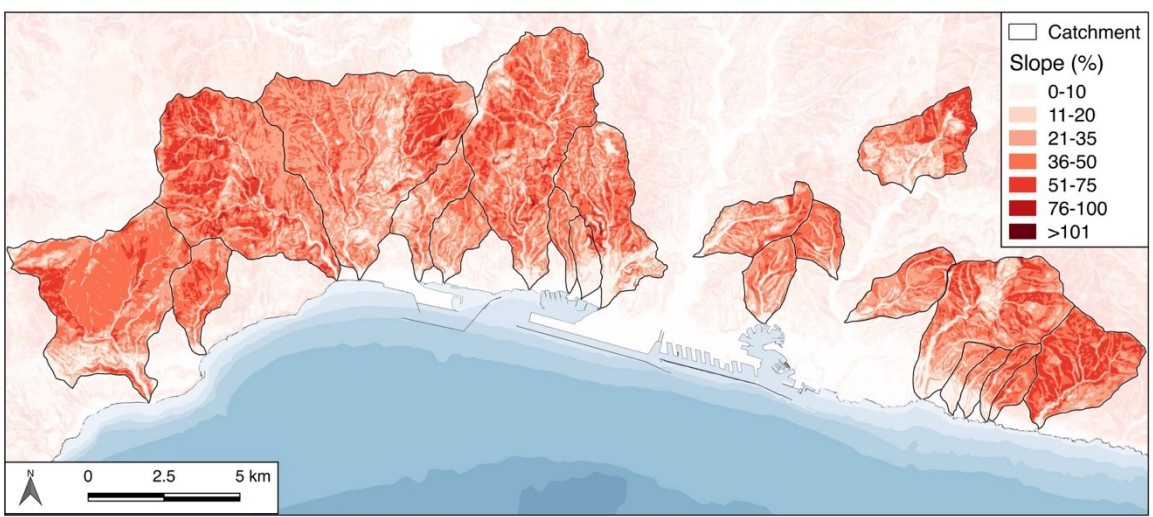

Figure 2:


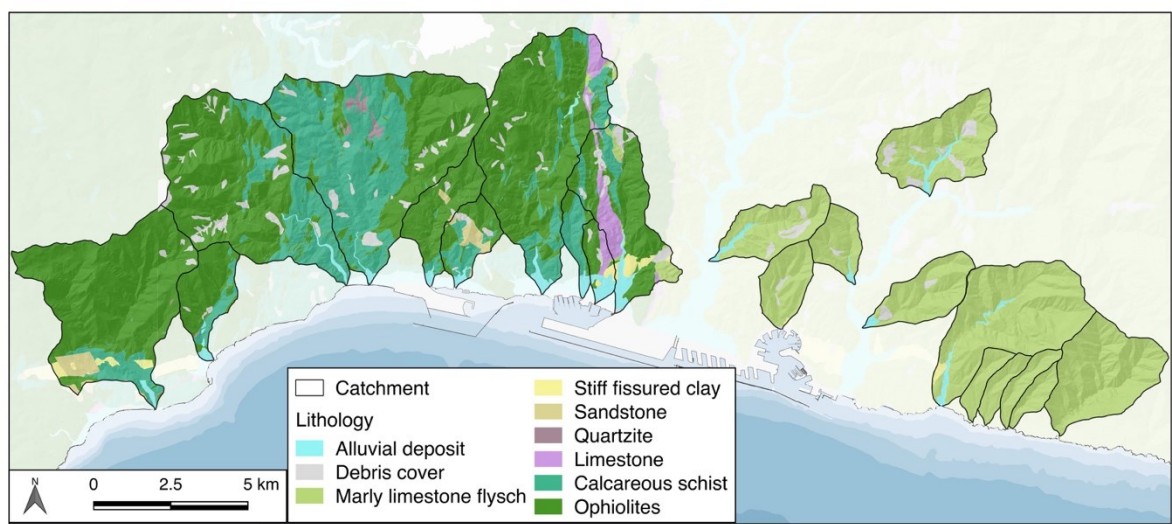

Figure 3:

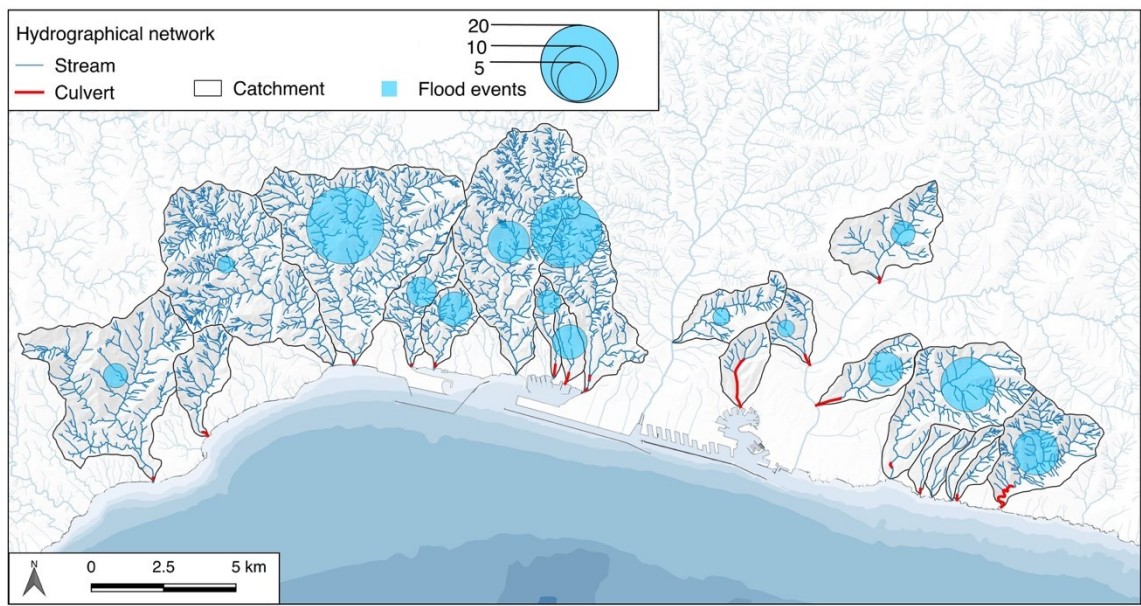


Figure 4:

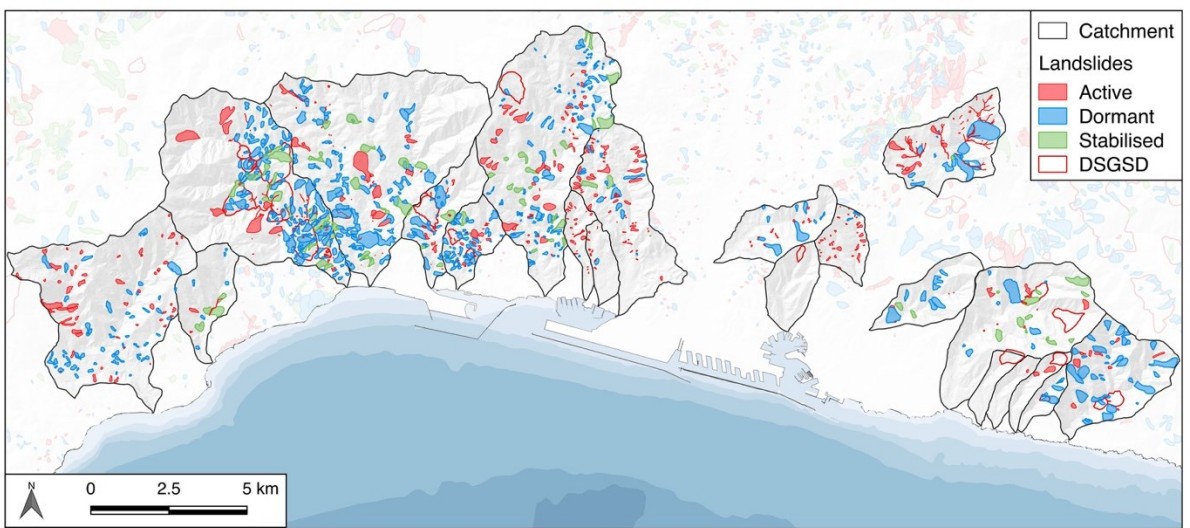

Figure 5:


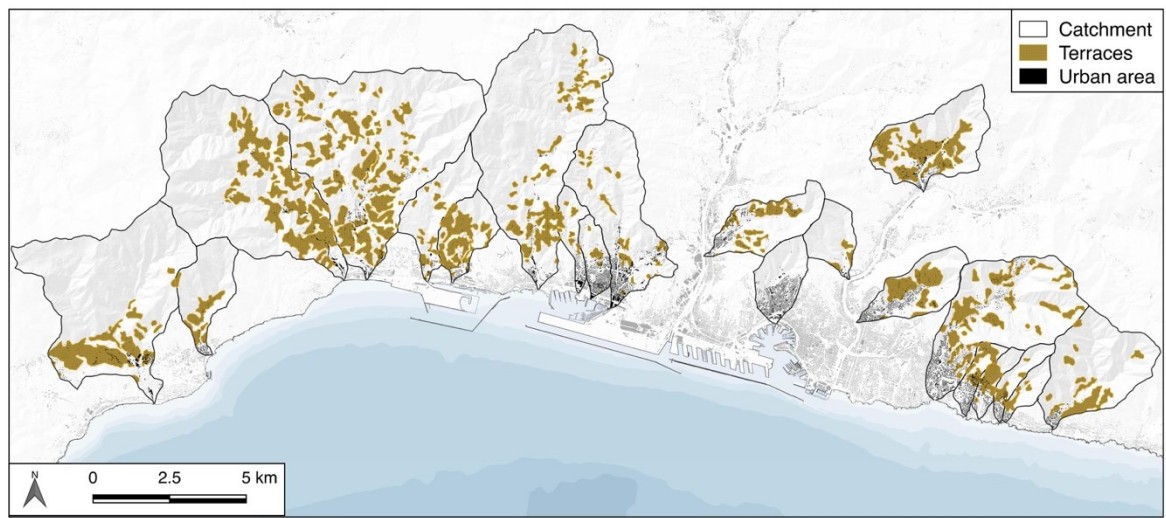

Figure 6:

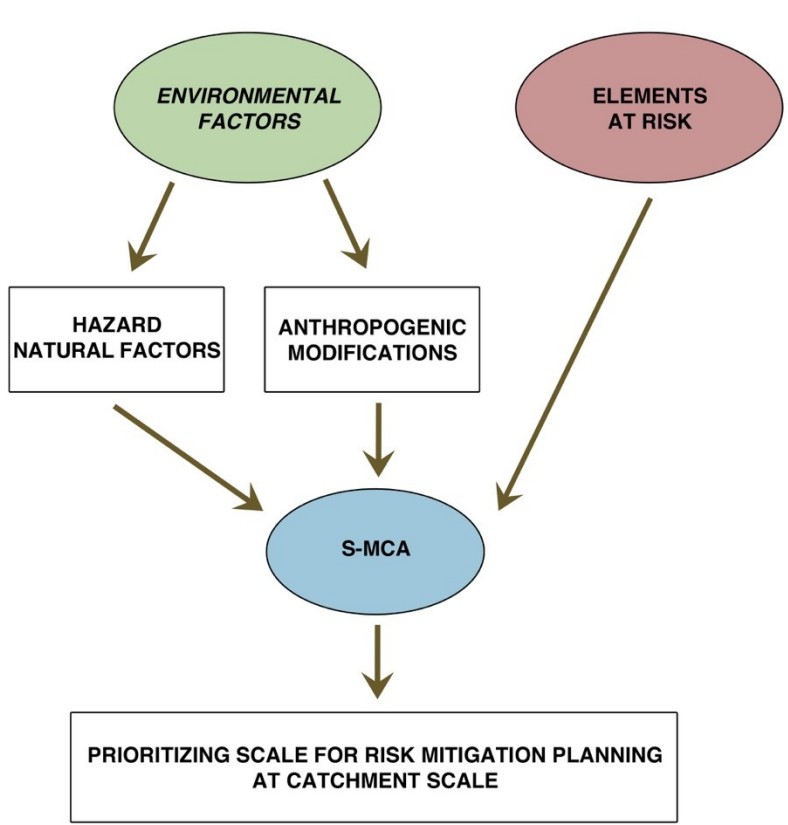


Figure 7:

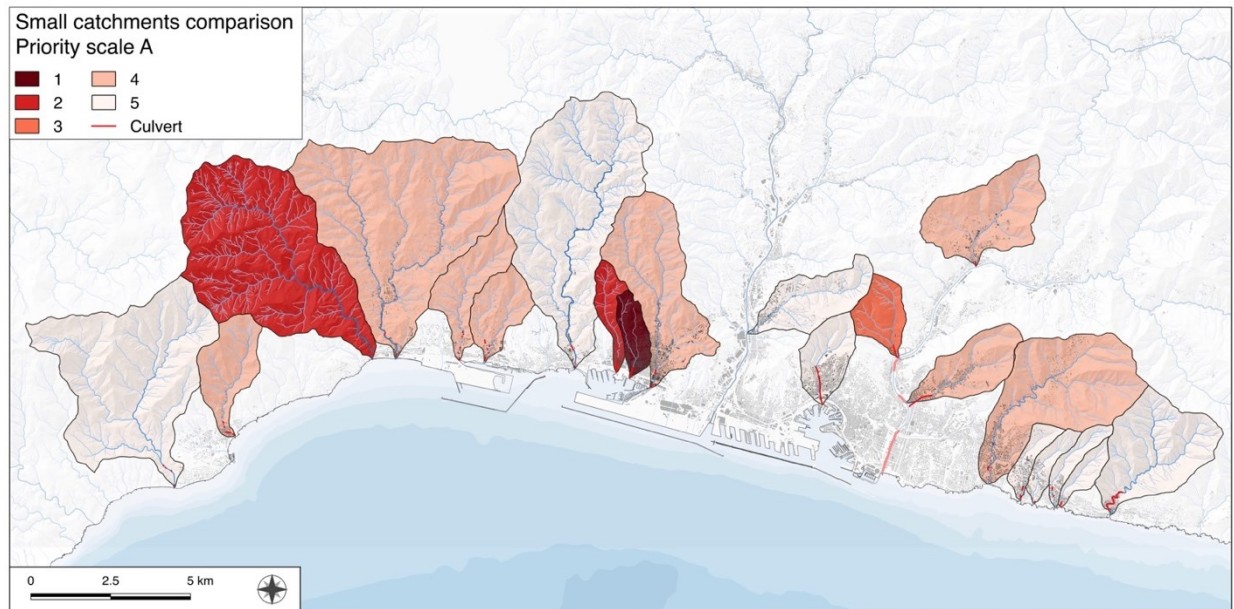

Figure 8:

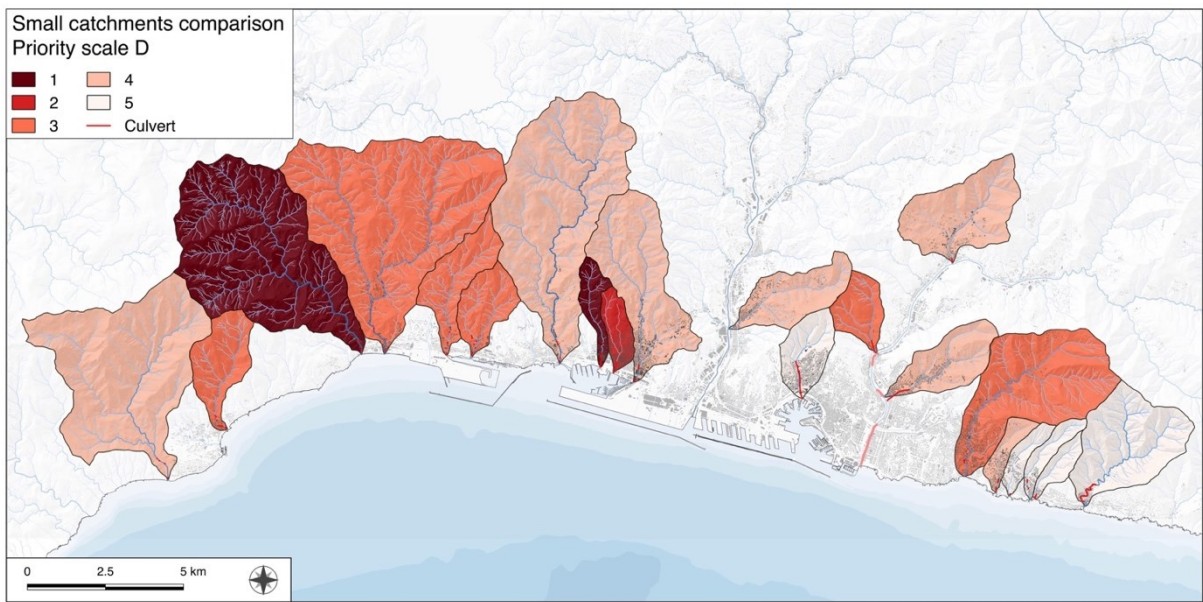

Figure 9: