# Peer review of "A spatial multicriteria prioritizing approach for geo-hydrological risk mitigation planning in small and densely urbanized Mediterranean basins"

_Natural Hazards and Earth System Sciences, 2018_

## Referee Comment (RC1) · Anonymous Referee #1 · 13 Jul 2018

**Review of the paper "A spatial multicriteria prioritizing approach for geohydrological risk mitigation planning in small and densely urbanized Mediterranean basins" submitted to NHESS by Guido Paliaga, Francesco Faccini, Fabio Luino, Laura Turconi**

**General Comments**

Following the proposal made by the authors the article deals with the proposal of three groups of indicators to constitute a priority scale to support the long-time planning interventions at catchment scale. The authors propose 18[th] indicators that group in: environmental factors-natural evolution (set 1); environmental factors-anthropogenic impact (set 2); social factors (set 3). They obtain this information for 21 little catchments of the Liguria Region and apply the multicriteria analysis technique. The authors include a long list of references to support their paper.

Although the topic of the paper is interesting and could be applied to other similar catchments in Mediterranean Region the paper lacks of information about the selected criteria and methodology. On the contrary, the paper seems a technical report for local policymakers, giving so much detail for each basin that is not necessary in an international publication and does the reading of the paper difficult. It is not necessary to detail the features and results for each basin. You could select the most relevant ones as example of the methodology.

In conclusion, in my opinion, this paper deserves publication in NHESS after major changes. Please, take into account my previous comments and the following ones. I would recommend that English language would be reviewed by an English native translator.

**Abstract:**

The abstract needs to be improved, both from the content and the redaction (i.e. "giving a **support** tool for decision makers, **supporting** a strong scheduling". I would recommend including more specific information about the region of study, the database and methodology as well as results. On the contrary, the first introductory paragraph (Lines 17-33) could by shortened and the last one (Lines 32-35) should be modified because it does not transmit a clear message. Why do you say "obtaining the optimization of economic resources"? I have not seen any economic analysis, neither the relationship of this analysis with the three set of parameters.

**Introduction**

Please, make a deep review of the Introduction. For instance, you say three times practically the same: "due to particular characteristics of geology, geomorphology and climate that can induce a high geo-hydrological hazard" (Page1, lines 40-42); "the general climatic context, with the interface between cold air masses and the sea, a steep territory and a complex geomorphologic and geologic context are the main natural factors" (Page 2, lines 53-55); "the general climatic context, with the interface between cold air masses and the sea, a steep territory and a complex geomorphologic and geologic context are the main natural factors" (Page 2, lines 59-61).

On the other hand Mediterranean region is the interface between cold air masses from the North (Atlantic or Continental) and warm subtropical and tropical air masses. The role developed by

the sea varies along the year, but the most important is the strong potential instability at low levels that characterize the Mediterranean air mass, as well the high water vapour content.

The paragraphs included from line 40 to line 93 show a general introduction about the Mediterranean region, and flood and landslides hazards. This is not bad; however, some references to other scientific works performed with spatial multicriteria analysis or dealing with support tools to plan long-term interventions at catchment scale, should be included in the Introduction, in order to know the state of the art.

**Data and methodology:**

Some aspects of the methodology deserve clearer and more elaborate explanation:

- Which is the meaning of the acronyms "IFFI, AVI, CTR, DSGDS,…"? Which is the source of the flood hazard map? Ad the source of all the information used in the paper.
- Which period do you use for the "Flood data from the AVI archive"? (write "flood data", not "floods data")
- Where are included social data (population density, economy data,…)?
- How do you characterize the risk level? It does not appear in the paper. Please, explain it.
- Why you have selected these indicators? Have you published a previous work with them? Is there any literature about it?
- The most important contribution for the scientific community would be the parameters selection and the multicriteria methodology. However, the only information that appears about them is the list of parameters and that "the S-MCA has been performed through the geo-UmbriaSUIT plugin available in Quantum GIS software, and the software performs a TOPSIS (Technique for Order of Preference by Similarity to Ideal Solution) multicriteria method (Huang and Yoong, 1981)". But, which is the philosophy of this methodology? How do you justify the classification showed in Table 7? Is this software free for all the public? The reference is old; do you have any more recent reference about this methodology? How do you rank the priorities?
- Which kind of survey have you made? Which was the target people?

**Discussion**

As I have proposed in the General Comments it would be interesting to select some catchments as example to show the methodology and to discuss the potential solutions (structural and non-structural) that could be adopted for each one.

Which mitigation works would be proposed depending of the scale of priorities showed in Table 7? It would be interesting to introduce a figure or a table showing the classification of priorities, the indicators or set of indicators that each priority considers and the potential solutions that could be applied. Discussion could consider if they are urban catchments or not, economic and ecological limitations, or the potential acceptation of the population

**Minor changes:**

Page 1, line 19: Authors say "The high hazard is often associated to intense urbanization…" but urbanization also affects vulnerability and exposure. Please, substitute the term "hazard" by "risk".

Page 4, line 154. Add a parenthesis to "fig. 3)"

Page 4, line 156. I think that "present both the lithology" should be "present both lithologies".

Table 6: The caption of the table says that "using parameters k, l 674 and m (ref. Tab. 2);", but they do not appear in Table 2. The same with "parameters a through j".

Page 5, lines 192-194: You say "due to the Mediterranean cyclones that periodically spring and intensify from south of the Alps over the Gulf of Genoa in the Ligurian Sea". In spite that this phenomenon is correct, usually and due to the orography of the region, there are a great part of the events that comes from the Mediterranean (with or without a cyclone in surface). The main cyclogenesis is over the sea on the Gulf of Genoa.

Page 5, line 205. Please, add a reference to justify these values.

Page 6, line 220. Please, substitute "their" by "its in the text: "because of their contributing effect to risk".

---

## Referee Comment (RC2) · Anonymous Referee #2 · 15 Jul 2018

The paper aims to propose a support tool to decision makers to plan and schedule long term investigations at catchment scale in the region of Liguria in northern Italy. Small catchments in a high hazard area have been assessed and compared through three sets of parameters: one describing the morphometric-morphological features related to flood and landslide hazard, another describing the degree of urbanization and of anthropogenic modifications at catchment scale and the last related to the elements at risk exposed. To address the main objective, multi criteria analysis technique to the descriptive parameters was applied.

General comment

I have read the paper with great interest and the main objective addressed by the manuscript is framed to the scope of the journal, but there are some confusions. My main concern is that the paper reflects more an engineering approach rather than a research approach. Therefore, I think that the paper needs some revisions and I recommend accepting it only after these revisions.

Specific comments

Introduction. In general, flood risk in the context of natural hazards is a broad term, which covers different dimensions from physical to social approaches. In this line, it is important from the authors to give a clear framework of the concept used in this study. Try to explain better or make more explicit the links what you deal with. In this part and to avoid confusion, I would suggest the authors to clearly indicate the flood processes in the area, to better define the problem and to explain better why used the described approach. To make the paper more relevant for the readers of this journal, I would suggest making a more explicit link to ongoing research in the natural hazard community.

Materials and Methods part.

The study area is well described. I would suggest the authors to reduce the information (parts: Geomorphological and geological settings and Climate and Meteorological context) by focusing only on important info for this study. The methodological outline is good described, and the method sounds scientifically correct (I am not an expert on statistics).

In page 7/line 273 where the data is described, the authors used a DEM realized in 2007 and a land use dataset realized in 2015. I would suggest them to use a newer elevation model and if it possible a DTM rather a DEM to reduce uncertainty on their simulations. Moreover, I would suggest them to add units of the formulae parameters

used on Table 2 and Table 3 to avoid confusion, to explain some abbreviations used and to describe more the survey performed. Additionally, and as authors used the International System of Units (SI) I would suggest them to check if the formulas used are in this system. On Table 3 (NRCS-SCS Line) the formula presented is in inches and they are dealing with millimeters. Moreover, it is not entire clear to me, how do they calculate the areas exposed to risk level R1-R4.

Results/Discussion.

In general, I would suggest the authors to merge these parts and to discuss their findings based on the methodology used and/or findings from other similar studies. What is missing in my opinion is a connection or a comparison of their findings with the international literature and/or with findings form other case studies (In the discussion part is only on reference on other studies).

At the end, the conclusions presented are too general and do not reflect what exactly shown in this study. Conclusions based on the findings of the analysis presented would be more effective.

---

## Author Comment (AC1) · 24 Jul 2018

We wish to thank the reviewer for comments and for the proposed changes for improving the paper, but we would like to detail some aspects that maybe do not arise clearly from the manuscript. The manuscript is aiming to give a possible answer to risk mitigation planning in a high-risk area that in the past, recent and not, has been hit by flood events and numerous rains induced shallow landslides, causing casualties and high damage. The more recent one (2014) caused 1 fatality and about 200 mln. € cost. Heavy rain events, characterized by a strong localization, have hit some of the

small catchments but not others that, in the future, could be affected. Then we faced the problem of comparing the features of several small catchments in order to address risk mitigation strategies and planning in terms of priority between the catchments in a densely populated area. In our experience the risk mitigation works that have been done in the area in the last 30 years have substantially failed or do not completely fulfill the target for two main reasons: the lack in a holistic approach, that is ignoring the catchment scale processes and interactions, and a wide strategy and planning of works. In other words, no adequate planning according to priorities at catchment scale, no adequate approach to the problem itself and high cost after every heavy rain event. Actually, some works are ongoing in some catchment approaching only the hydraulic factor and ignoring the concurrent contribution of debris flow and strong solid transport in streams that occurs in occasion of heavy rain events, saturating the capacity of final stretch culverted streams. Such kind of events happened in the past during the 2010 flood that hit Chiaravagna catchment (one of the studied ones and published by some of the authors), 2011 flood in 5 Terre and Vara valley, 2014 flood in the Veilino, Geirato and Cerusa catchment, both of them included in the performed analysis, and during the 1970 and 1953 again in the Geirato one. The combined effects of heavy rains, morphometric features that determine reduced time of concentration, rain induced shallow landslides and culverted streams in highly anthropogenic modified context, caused devasting effects. Then we focused our attention on natural factors that cause hazard and on the strong anthropogenic modifications that have magnified the hazard effects like the culverting of streams and soil consumption, which is related to the constriction and confining of streambed. (The natural factors parameters are the ones that describe the features of the territory in terms of possible floods and in terms of rains induced shallow landslides: time of concentration, mean slope etc., together with the ones that describe the availability of potentially destabilizing deposits on the slopes.) The identified parameters, in our opinion and experience and after the effects of the cited events, are the ones more suitable for the local conditions. Maybe some of them could be redundant or maybe others could be added, but the approach we propose

does not change: a methodology to realize of a catchments priority scale to identify the more urgent to act on. In different environmental conditions, different criteria could be more appropriate. A more rigorous approach maybe should have started calibrating the parameters on the effects after heavy rain in different catchments but we would need almost identical events hitting different catchments at the same time that is quite difficoult, due even to the strong morphological heterogeneity of the area. Besides the more recent events are strongly localized, then affecting a small catchment but not the neighboring one. Identifying the more critical catchments could allow to design the operative planning of works, after a detailed basin scale analysis. Relating inhomogeneous parameters like natural factors, anthropogenic modifications and social ones, related to the exposure to flood risk, has been done applying the spatial multi-criteria analysis techniques, which are widely used in planning, environmental impact assessment and strategic environmental assessment to compare several choices. The comparing technique is here proposed to realize a classification of the worst conditions.

---

## Author Comment (AC2) · 24 Jul 2018

We wish to thank the reviewer for comments and for the proposed changes for improving the paper. If the final reviewers and the editor will agree we will modify and integrate the manuscript according to the requests, trying to make better arise and clarify the proposed methodology that tries to give an answer to the risk mitigation strategies in a critical area through a priority scale classification of small catchments.

2018-100, 2018.

---

## Author Response (AR1)

**A spatial multicriteria prioritizing approach for geo-hydrological risk mitigation planning in small and densely urbanized Mediterranean basins**

Guido Paliaga[1], Francesco Faccini[2], Fabio Luino[1], Laura Turconi[1]

[1]CNR IRPI Research Institute for Geo-Hydrological Protection – Strada delle Cacce 73, 10135 Torino (Italy)
[2]UNIVERSITA' DI GENOVA – DISTAV Department of Earth, Environmental and Life Sciences, Genoa University (Italy)

**Author's response to the referee**

We wish to thank the referee as their comments helped us to improve the manuscript, give a we hope more clear explanation of our approach and to correct mistakes.

We think that the paper fits the aim of the journal, trying to address with a quantitative methodology the process of risk reduction strategies and then giving a decision support tools that could help to underline the worst catchment condition. The tool could be used even to monitor the progress of actions during the long time needed to realize in particular the structural interventions.

Finally, we think that, opportunely adapted, the method could be used in different context with diverse active processes and situations and even for a comparison of sub-catchments.

The paper has been reviewed by a native English speaker.

**Reviewer 1**

**Comment from the referee**

**Abstract:**

The abstract needs to be improved, both from the content and the redaction (i.e. "giving a **support** tool for decision makers, **supporting** a strong scheduling". I would recommend including more specific information about the region of study, the database and methodology as well as results. On the contrary, the first introductory paragraph (Lines 17-33) could by shortened and the last one (Lines 32-35) should be modified because it does not transmit a clear message. Why do you say "obtaining the optimization of economic resources"? I have not seen any economic analysis, neither the relationship of this analysis with the three set of parameters.

**Response**

We agree with the request.

**Changes in the manuscript**

The Abstract has been fully revised and rewritten: more details have been included and the philosophy of catchments comparison is more clearly declared.

**Comment from the referee**

**Introduction**

Please, make a deep review of the Introduction. For instance, you say three times practically the same: "due to particular characteristics of geology, geomorphology and climate that can induce a high geo-hydrological hazard" (Page1, lines 40-42); "the general climatic context, with the interface between cold air masses and the sea, a steep territory and a complex geomorphologic and geologic context are the main natural factors" (Page 2, lines 53-55); "the general climatic context, with the interface between cold air masses and the sea, a steep territory and a complex geomorphologic and geologic context are the main natural factors" (Page 2, lines 59-61).

On the other hand Mediterranean region is the interface between cold air masses from the North (Atlantic or Continental) and warm subtropical and tropical air masses. The role developed by
the sea varies along the year, but the most important is the strong potential instability at low levels that characterize the Mediterranean air mass, as well the high water vapour content.

The paragraphs included from line 40 to line 93 show a general introduction about the Mediterranean region, and flood and landslides hazards. This is not bad; however, some references to other scientific works performed with spatial multicriteria analysis or dealing with support tools to plan long-term interventions at catchment scale, should be included in the Introduction, in order to know the state of the art.

**Response**

We agree with the request.

**Changes in the manuscript**

The introduction has been deeply reviewed and rewritten according to the reviewer's comments and requests.

References for the spatial multicriteria analysis has been integrated with more recent works, giving a wider state of the art view.

**Comment from the referee**

**Data and methodology:**

Some aspects of the methodology deserve clearer and more elaborate explanation:

1. Which is the meaning of the acronyms "IFFI, AVI, CTR, DSGDS,..."? Which is the source of the flood hazard map? Add the source of all the information used in the paper.
2. Which period do you use for the "Flood data from the AVI archive"? (write "flood data", not "floods data")
3. Where are included social data (population density, economy data,...)?
4. How do you characterize the risk level? It does not appear in the paper. Please, explain it.
5. Why you have selected these indicators? Have you published a previous work with them? Is there any literature about it?
6. The most important contribution for the scientific community would be the parameters selection and the multicriteria methodology. However, the only information that appears about them is the list of parameters and that "the S-MCA has been performed through the geo-UmbriaSUIT plugin available in Quantum GIS software, and the software performs a TOPSIS (Technique for Order of Preference by Similarity to Ideal Solution) multicriteria method (Huang and Yoong, 1981)". But, which is the philosophy of this methodology? How do you justify the classification showed in Table 7? Is this software free for all the public? The reference is old; do you have any more recent reference about this methodology? How do you rank the priorities?
7. Which kind of survey have you made? Which was the target people?

**Response**

We agree with the request.

**Changes in the manuscript**

1. All the acronyms in the paper have been displayed; the source of all the data, comprising flood hazard maps, is the regional authority – Regione Liguria, as was written before the list of the data (line 272) but it is now more explicit.
2. The period, as it comes from the two databases, is 1900-1990 and 2005-2016.
3. / 4. We have explained not clearly and maybe misusing the term: social data is used in S-MCA when applied to project comparison; in our application of the methodology we used for it the exposed elements (areal and punctual) that the Regional authority has used in the application of the EU flood directive: buildings, residential areas, hospitals, schools, cultural heritages area considered when present in the flood hazard zones. As we explain in the corrected text, exposure at risk is defined considering the elements that may be threatened by floods as they have been adopted by the local authority- Regione Liguria - after the hydraulic modeling, that is the hazard assessment, and the evaluation of the potential damages, then vulnerability. The official data define areas and punctual elements exposed to 4 increasing risk levels from R1 to R4.
4.
5. The indicators have been selected during the field survey basing on the evaluation of the geomorphic active processes in the area related to geo-hydrological hazard and on the effects caused by previous intense rain events. Then we used previous papers (Cevasco et al., 2017; Giordan et al. 2017; Faccini et

al, 2018) and the effects of the recent events in 2010, 2011, 2014 and partially 2015. As we have written in the text the indicators are used to represents the situation but in others context they may change depending on the different peculiarities. Where abandoned terraces are not present, that indicator would not be necessary but others may substitute it.

6. We give further information about TOPSIS methodology with more recent references; both the plugin and Quantum Gis software are free. The explanation of the method and of the approach we used, is more clearly explained and is based on the comparison of heterogeneous features of elements, in order to realize a ranking that, in our case, is representative of the degree of attention that should be used in planning the risk reduction activities. The comparison, and then the ranking, is between catchments, trying to underline the necessity of acting at catchment scale and not only considering some few interventions as the only and final ones.

7. A field survey was performed on the studied catchments, evaluating slope stability, possible sources of debris/mud flows, hydrographical network conditions, comprising the artificially modified ones, the extension and typology of the areas and elements present in the flood hazard zones.

**Comment from the referee**
**Discussion**

As I have proposed in the General Comments it would be interesting to select some catchments as example to show the methodology and to discuss the potential solutions (structural and non- structural) that could be adopted for each one.

Which mitigation works would be proposed depending of the scale of priorities showed in Table 7? It would be interesting to introduce a figure or a table showing the classification of priorities, the indicators or set of indicators that each priority considers and the potential solutions that could be applied. Discussion could consider if they are urban catchments or not, economic and ecological limitations, or the potential acceptation of the population.

**Response**

We agree with the request.

**Changes in the manuscript**

We have included in the text what kind of prevention activities should be adopted to reduce the high risk in the area. All the catchments are urban, as described in the paragraph 2.1: as emerge from the fig.1, all the catchments present a more or less extended natural zone in the higher parts and a strongly urbanized one in the lower parts. The parameter k, soil consumption and l, culvert last km, were used to describe that variability.

**Comment from the referee**
**Minor changes:**

1. Page 1, line 19: Authors say "The high hazard is often associated to intense urbanization..." but urbanization also affects vulnerability and exposure. Please, substitute the term "hazard" by "risk".
2. Page 4, line 154. Add a parenthesis to "fig. 3)"
3. Page 4, line 156. I think that "present both the lithology" should be "present both lithologies".
4. Table 6: The caption of the table says that "using parameters k, l 674 and m (ref. Tab. 2);", but they do not appear in Table 2. The same with "parameters a through j".
5. Page 5, lines 192-194: You say "due to the Mediterranean cyclones that periodically spring and intensify from south of the Alps over the Gulf of Genoa in the Ligurian Sea". In spite that this phenomenon is correct, usually and due to the orography of the region, there are a great part of the events that comes from the Mediterranean (with or without a cyclone in surface). The main cyclogenesis is over the sea on the Gulf of Genoa.
6. Page 5, line 205. Please, add a reference to justify these values.
7. Page 6, line 220. Please, substitute "their" by "its in the text: "because of their contributing effect to risk".

**Response**

We agree with the request.

**Changes in the manuscript**

1. The sentence has been re-written.
2. It has been corrected.
3. It has been corrected.
4. There was an error in the caption: the right reference is to table 4.
5. We have corrected the sentence coherently to the referee request.
6. References have been added.
7. It has been corrected.

**Reviewer 2**

**Comment from the referee**

**Introduction.**

In general, flood risk in the context of natural hazards is a broad term, which covers different dimensions from physical to social approaches. In this line, it is important from the authors to give a clear framework of the concept used in this study. Try to explain better or make more explicit the links what you deal with. In this part and to avoid confusion, I would suggest the authors to clearly indicate the flood processes in the area, to better define the problem and to explain better why used the described approach. To make the paper more relevant for the readers of this journal, I would suggest making a more explicit link to ongoing research in the natural hazard community.

**Response**

We agree with the request.

**Changes in the manuscript**

Introduction has been completely re-written according even to the comments of the reviewer 1; we think we have more clearly described flash flood processes in the area and the concurrent shallow landslides that are activated during intense rain events.

**Comment from the referee**

**Materials and Methods part.**

The study area is well described. I would suggest the authors to reduce the information (parts: Geomorphological and geological settings and Climate and Meteorological context) by focusing only on important info for this study. The methodological outline is good described, and the method sounds scientifically correct (I am not an expert on statistics).

In page 7/line 273 where the data is described, the authors used a DEM realized in 2007 and a land use dataset realized in 2015. I would suggest them to use a newer elevation model and if it possible a DTM rather a DEM to reduce uncertainty on their simulations. Moreover, I would suggest them to add units of the formulae parameters used on Table 2 and Table 3 to avoid confusion, to explain some abbreviations used and to describe more the survey performed. Additionally, and as authors used the International System of Units (SI) I would suggest them to check if the formulas used are in this system. On Table 3 (NRCS-SCS Line) the formula presented is in inches and they are dealing with millimeters. Moreover, it is not entire clear to me, how do they calculate the areas exposed to risk level R1-R4.

**Response**

We partially agree with the request.

**Changes in the manuscript**

Some information related to the study area have been reduced.

The DEM acronym was a typing error: we used a DTM from the regional authority, the more recent one acquired in 2007.

We added all the units in the formulae and included the number of streams of order u in tab. 2 (nu) whose lack could generate ambiguities. The NRCS-SCS formula we used, was in SI units, as the imperial system one is the following:

$$t_c = 0.0526 * \frac{L_m^{0.8} * (X + 1)^{0.7}}{y^{1/2}}$$

Where: $[t_c]$ = hours, $[L_m]$ = km and $[y]$ = %

As we explain in the corrected text, exposure at risk is defined considering the elements that may be threatened by floods as they have been adopted by the local authority- Regione Liguria - after the hydraulic modeling, that is the hazard assessment, and the evaluation of the potential damages, then vulnerability. The official data define areas and punctual elements exposed to 4 increasing risk levels from R1 to R4 and comprises residential areas, schools, cultural heritages, hospitals.

**Comment from the referee**

    **Results/Discussion.**

    In general, I would suggest the authors to merge these parts and to discuss their findings based on the methodology used and/or findings from other similar studies. What is missing in my opinion is a connection or a comparison of their findings with the international literature and/or with findings form other case studies (In the discussion part is only on reference on other studies).

    At the end, the conclusions presented are too general and do not reflect what exactly shown in this study. Conclusions based on the findings of the analysis presented would be more effective.

**Response**

    We partially agree with the request.

**Changes in the manuscript**

    Trying to address both the request of reviewer 1 and 2 we left the division between Result and discussion but we integrated them with a more detailed discussion and eliminating specific comments on the various catchments in the result.

    We integrated references on other studies and detailed the interventions that could be done to solve the problems that emerged as result from the performed analysis.

**A spatial multicriteria prioritizing approach for geo-hydrological risk mitigation planning in small and densely urbanized Mediterranean basins**

25  Guido Paliaga[1], Francesco Faccini[2], Fabio Luino[1], Laura Turconi[1]

[1]CNR IRPI Research Institute for Geo-Hydrological Protection – Strada delle Cacce 73, 10135 Torino (Italy)
[2]UNIVERSITA' DI GENOVA – DISTAV Department of Earth, Environmental and Life Sciences, Genoa University (Italy)

*Correspondence to*: Fabio Luino (fabio.luino@irpi.cnr.it)

**Abstract**

35  Landslides and floods, particularly flash floods, occurred currently in many Mediterranean catchments as a consequence of heavy rainfall events, causing damage and sometimes casualties. The high hazard is often associated with high vulnerability deriving from an intense urbanization in particular along the coastline where streams are habitually culverted. The necessary risk mitigation strategies should be applied at catchment scale with a holistic approach, avoiding spot interventions.

40  In the present work a high-risk area, hit in the past by several floods and concurrent superficial landslides due to extremely localized and intense rain events, has been studied. 21 small catchments have been identified: only some of them have been hit by extremely damaging past events, but all lies in the intense rain high hazard area and are strongly urbanized in the lower coastal zone. The question is what would happen if an intense rain event should stroke one of the not previously hit catchment; some situations could be worse or not, so the attention has been focused on the comparison between catchments. The aim

45  of the research has been identifying a priority scale between catchments, pointing out the more critical ones and giving a quantitative comparison tool for decision makers to support a strong scheduling of long-time planning interventions at catchment scale. The past events effects and the geomorphic processes analysis together with the field survey allowed to select three sets of parameters: one describing the

50  morphometric-morphological features related to flood and landslide hazard, another describing the degree of urbanization and of anthropogenic modifications at catchment scale and the last related to the elements

that are exposed to risk. The realized geodatabase allowed to apply the spatial multicriteria analysis technique (S-MCA) to the descriptive parameters and to get to a priority scale between the analyzed catchments. The scale can be used to plan risk mitigation interventions starting from the more critical catchments, then focusing economic resources primarily on them and obtaining an effective prevention strategy. The methodology could be useful even to check how the priority scale is modified during the progress of the mitigation works realization.

Besides, this approach could be applied in similar context, even between sub-catchments, after identifying a suitable set of descriptive parameters depending on the active geomorphological processes and the kind of anthropogenic modification. The prioritization would allow to invest economic resources in risk mitigation interventions priory in the more critical catchments.

Landslides and floods, particularly flash floods, occurred currently in many Mediterranean catchments as a consequence of heavy rainfall events, causing damages and sometimes casualties. The high hazard is often associated to intense urbanization in particular along the coastline where streams are habitually culverted. The necessary risk mitigation strategies should be applied at catchment scale, considering the concurrent landslides and flood events and would need to be accurately planned in order to optimize the available economic resources.

In the present work 21 small catchments in a high hazard area have been assessed and compared through three sets of parameters: one describing the morphometric-morphological features related to flood and landslide hazard, another describing the degree of urbanization and of anthropogenic modifications at catchment scale and the last related to the elements that are exposed to risk. The aim of the research is to constitute a priority scale among the small catchments, applying the multicriteria analysis technique to the descriptive parameters and giving a support tool for decision makers, supporting 
[revised manuscript text omitted]

Anthropogenic modifications are interesting most of the coastal floodplain with a strong soil consumption and with the widespread narrowing of the streambed that has favored most of the flood prone areas (Faccini et al. 2015, Faccini et al. 2016). Furthermore, the modifications are often interesting even the hinterland: besides the urban sprawl and the fragmentation caused by infrastructures, in some areas the ancient man-made terraces realized for agricultural practice and actually largely abandoned, constitute an increasing factor of geomorphological hazard (Brancucci and Paliaga, 2006; Tarolli et al., 2014; Paliaga, 2016). In the recent years many evidences have been arising in Italy: large areas of Liguria (Brandolini

et al., 2017; Cevasco et al., 2017) and Toscana (Bazzoffi and Gardin, 2011) are interested by terraces instability that may turn in source of geomorphologic hazard. In the Mediterranean region many areas present similar occurrence of terraces with analogous problems: the French Côte d'Azur, the Mediterranean and insular Spain and Greece (Tarolli et al., 2014) are some example. In the recent years some disastrous events involved terraced slopes: in 2011, during the Cinque Terre flood (Liguria, northern Italy) (Brandolini et al., 2017; Luino and Turconi, 2017), many terraces collapsed and the subsequent debris filled villages at a height of about 3 m, and in 2014, in the Leivi village during the Chiavari flood (Liguria) a terraced slope collapsed destroying a house and causing 2 fatalities (Faccini et al., 2017; Luino and Turconi, 2017).

Every year in Italy many casualties and significant damages, both to private and public sectors, are caused by floods, landslides and debris flow: the 2017 periodic CNR-IRPI report (CNR, 2018) on Italian population landslides and floods threat, evidences 1789 casualties and 317.526 homeless in the period 1967-2016, with all the regions affected. Liguria, despite its small surface, is between the most affected region scoring the third place in the mortality index calculated on both landslide and flood events.

Risk mitigation strategies are more and more urgent but largely disregarded, unapplied or only partially pursued: few resources are allocated and, commonly, are used only for emergency actions while a long-term planning and scheduling should be crucial to obtain significant results (Prenger-Berninghoff et al., 2014). In the recent years, in Italy, some large structural works have been started to mitigate the worst flooding risk situations, but without following a broad approach at catchment scale. The most important is the floodway channel for the Bisagno stream in Genoa (Liguria), but similar project or culvert adjusting are ongoing in smaller neighboring streams. This approach allows to mitigate just a part of the risk, ignoring slope instability processes and related contribution to solid transport into hydrographical network.

In Liguria, and especially in the Genoa metropolitan area, the high geo-hydrological risk is related to intense urbanization, to the geomorphological asset and to heavy rainfall that is generally intense in autumn (Silvestro et al. 2012) and that appears to be increasing in intensity (Faccini et al., 2015; Aquaotta et al., 2017). Apart the structural interventions in the larger Bisagno catchment, even the smaller ones in

215

~~The aim of the research is to propose a support tool to decision makers in order to plan and schedule long-term interventions at catchment scale. A group of 21 small catchments in the middle of the zone more liable to heavy rainfall (Cassola et al., 2016) have been analyzed, comparing three sets of descriptive parameters. The comparison has been performed with spatial multicriteria analysis using a total of 19~~
220 ~~parameters and obtaining a priority scale between the 21 catchments. Spatial multicriteria approach has been applied by many authors in flood risk and in natural hazard management (Gamper et al., 2006; de Brito et al., 2006), mostly to identify specific areas prone to instability or to compare catchments through morphometric parameters (Benzougagh et al., 2017). In the present work the Authors applied those techniques considering a broader set of parameters, trying to address the peculiarity of small urban~~
225

**2 Material and method**

230 *2.1 Geomorphological and geological settings*

The studied area is one of the most critical in terms of geo-hydrological risk in Italy and in the Mediterranean basin, due to the morphometric features and to the high urbanization. It is located in the central part of Liguria region, northern Italy (fig.1): 21 catchments with a surface area comprised between 1.3 and 27.5 km$^2$ have been analyzed. Five of them, numbered 11, 13, 14 and 15 in fig. 1, are sub-
235 catchments of the two major ones that cross Genoa city: the Bisagno and Polcevera catchments. The confluence of n° 13 with Polcevera is just upward the already collapsed Morandi bridge. All the others flow directly into the Ligurian sea.

The area is densely populated, 2,429 inhab/km$^2$ in the whole Genoa administration unit (ISTAT, 2012) and has been strongly urbanized starting from the beginning of the 20$^{th}$ century (Faccini et al., 2016).

240 Land use (fig. 1) clearly shows the strong dualism between the urban area, mainly concentrated in the lower catchments close to the sea, and the middle and upper mountainous catchments that preserve natural features with meadows and woods. Some catchments have been strongly modified by urbanization: in particular n° 8, 9, 10, 12, 15 and 16. In the upper part of catchments 11,12 and 13 the natural features and the presence of cultural heritages is testified by a highly frequented urban park. (Sacchini et al, 2018).In

245 the upper portion of n° 10 is present, since more than 50 years, the city dump and, in the central portion, some limestone quarries.

Neotectonics activity has deeply influenced the structural asset, catchments' morphometry and hydrographical network features (Paliaga, 2015). The catchments are mainly elongated and oriented orthogonally to the coast line and reach maximum altitudes comprised between 491 and 1189 m a.s.l.

250 (tab. 1). Only n° 1, 3 and 4 present a less elongated feature. The strong steepness of the slopes and a substantial lack of coastal floodplain is a distinctive feature of all the area: slope gradient is high in all the catchments and particularly in n° 3 and 21 (fig. 2). The only relatively extended floodplains are present in catchments n° 8, 9, 10, 14 and 16.

The catchments present substantial homogeneous lithological features if considered in three groups (fig.

255 3): the western one (from n° 1 to 7) are prevalently ophiolitic and metamorphic; the eastern (from n° 11 to 21) are essentially sedimentary, while the central ones (from n° 8 to 10) present both the lithologyies. This structure corresponds to the limit between, respectively, Alpine and Apennine structural units, with the Sestri-Voltaggio unit limit.

[revised manuscript text omitted]
° 9. For the set 1 parameters, drainage density, being affected by the bedrock and permeability, evidences higher values for the catchments from n°1 to 8, while mean slope is always high apart, relatively, for the smaller catchments n° 9, 17 and 18. Melton ratio is particularly high for catchments n° 19 and 20; ruggedness number presents high values for n° 3 and 7. Hypsometric index is always high due to the similar morphological asset of the catchments. Landslides surface, in percentage of the catchment surface, shows a large variability with the maximum values in catchments n° 3, 4, 5 and 6. Mean bifurcation ratio is low only for catchment n° 9. Time of concentration values are always short, due to the small dimensions of catchments and morphometric features; n° 5, 9, 17, 18, 19 and 20 have values shorter than 30 min. The higher number of flood events, more than 17, interested catchments n° 4, 10 and 16, while floodable areas shows a significant high value for catchment n° 9.

For set 2 parameters, soil consumption is always high and superior to 20%, at catchment scale, in n° 4, 5, 8, 9, 10, 12, 17 and 18; n° 9 reaches a value of about 50%. The final km culverted percentage for the main stream is over 30%, that means a 300 m long culvert, for n° 8, 9, 12, 15 and 21 with 100% for n° 12 and 21. Terraces are spread on more than the 20% of the surface in catchments n° 4, 6, 14, 15, 17 and 18.

Set 3 parameters describe the elements exposed at risk: catchments n° 8, 9 and 16 appears as the ones in the worst condition.

[revised manuscript text omitted]

Dd
(km⁻¹) | b
Mean gradient (%) | c
Mi | d
Rn | e
Hi | f
Landslide (%) | g
Rb mean | h
Time of concentration (') | i
Floods number | j
 Flood hazard  area 200 y (%) | k
Soil consumption (%) | l
Culvert last km (%) | m
Terraces (%) |
|---|---|---|---|---|---|---|---|---|---|---|---|---|---|
| 1 | 3.75 | 56.5 | 0.26 | 4.45 | 0.43 | 0.2 | 0.28 | 82.64 | 2 | 0.3 | 10.9 | 5.1 | 12.1 |
| 2 | 4.96 | 55.9 | 0.43 | 4.58 | 0.48 | 0.4 | 0.25 | 36.15 | 0 | 0.6 | 17.1 | 25.0 | 14.1 |
| 3 | 6.19 | 60.4 | 0.25 | 7.29 | 0.43 | 6.6 | 0.31 | 81.34 | 1 | 0.7 | 7.4 | 7.4 | 19.2 |
| 4 | 5.25 | 62.1 | 0.19 | 5.26 | 0.41 | 4.3 | 0.32 | 73.37 | 20 | 0.2 | 20.7 | 10.5 | 20.1 |
| 5 | 5.71 | 46.1 | 0.40 | 4.90 | 0.34 | 6.5 | 0.24 | 29.53 | 3 | 0.6 | 27.8 | 9.0 | 9.9 |
| 6 | 5.34 | 45.4 | 0.32 | 3.19 | 0.32 | 4.9 | 0.26 | 35.29 | 4 | 0.5 | 9.5 | 22.2 | 40.3 |
| 7 | 6.30 | 56.0 | 0.21 | 6.27 | 0.46 | 0.6 | 0.30 | 110.08 | 6 | 0.3 | 16.9 | 11.4 | 9.6 |
| 8 | 5.06 | 47.2 | 0.40 | 2.76 | 0.41 | 0.0 | 0.21 | 33.80 | 2 | 3.4 | 20.4 | 45.9 | 18.8 |
| 9 | 3.01 | 31.8 | 0.32 | 1.31 | 0.30 | 0.0 | 0.07 | 27.16 | 4 | 10.6 | 49.4 | 34.4 | 6.6 |
| 10 | 5.65 | 49.4 | 0.20 | 3.72 | 0.41 | 0.1 | 0.29 | 77.65 | 17 | 2.7 | 23.4 | 17.6 | 5.3 |
| 11 | 4.33 | 46.3 | 0.28 | 2.69 | 0.35 | 0.8 | 0.29 | 39.58 | 1 | 1.9 | 13.6 | 0.0 | 18.0 |
| 12 | 2.33 | 45.1 | 0.27 | 1.15 | 0.40 | 0.0 | 0.31 | 34.43 | 0 | 0.1 | 36.3 | 100.0 | 0.0 |
| 13 | 3.84 | 55.0 | 0.29 | 2.02 | 0.42 | 2.8 | 0.31 | 35.75 | 1 | 1.8 | 7.3 | 35.7 | 5.6 |
| 14 | 3.58 | 49.9 | 0.26 | 2.62 | 0.34 | 0.2 | 0.37 | 50.12 | 2 | 0.6 | 7.7 | 11.8 | 29.2 |
| 15 | 3.68 | 48.2 | 0.26 | 2.04 | 0.37 | 0.0 | 0.30 | 55.11 | 4 | 3.4 | 19.0 | 80.4 | 26.8 |
| 16 | 4.05 | 50.6 | 0.23 | 3.42 | 0.37 | 0.0 | 0.32 | 85.44 | 10 | 2.0 | 13.8 | 9.8 | 16.7 |
| 17 | 2.58 | 32.6 | 0.41 | 1.27 | 0.30 | 0.0 | 0.13 | 26.56 | 0 | 0.5 | 34.0 | 17.2 | 32.3 |
| 18 | 4.04 | 38.6 | 0.46 | 2.18 | 0.31 | 0.0 | 0.39 | 26.06 | 0 | 0.0 | 22.3 | 3.6 | 32.3 |
| 19 | 4.35 | 50.8 | 0.66 | 3.58 | 0.36 | 1.3 | 0.22 | 19.56 | 0 | 0.1 | 15.1 | 10.7 | 14.6 |
| 20 | 4.47 | 55.3 | 0.63 | 3.23 | 0.39 | 0.0 | 0.32 | 20.38 | 0 | 0.0 | 8.6 | 18.2 | 12.8 |
| 21 | 5.66 | 65.8 | 0.28 | 4.79 | 0.46 | 0.7 | 0.29 | 65.20 | 7 | 0.4 | 3.3 | 100.0 | 11.5 |

885

Table 5: The geodatabase with the evaluation of the surfaces (%) and punctual elements  to risk in the studied catchments, according to the EU Flood Directive 2007/60/CE.

| Catchment number | R1 risk area (%) | R2 risk area (%) | R3 risk area (%) | R4 risk area (%) | R2 risk elements | R4 risk elements |
|---|---|---|---|---|---|---|
| 1 | 0,19 | 0,02 | 0,00 | 0,16 | 0 | 0 |
| 2 | 0,14 | 0,61 | 0,02 | 0,48 | 1 | 0 |
| 3 | 0,16 | 0,18 | 0,04 | 0,53 | 6 | 0 |
| 4 | 0,08 | 0,18 | 0,00 | 0,14 | 2 | 1 |
| 5 | 0,04 | 0,41 | 0,01 | 0,38 | 1 | 1 |
| 6 | 0,07 | 1,03 | 0,00 | 0,45 | 0 | 0 |
| 7 | 0,13 | 0,37 | 0,00 | 0,17 | 1 | 0 |
| 8 | 0,20 | 2,30 | 0,00 | 3,21 | 0 | 2 |
| 9 | 0,02 | 0,24 | 0,00 | 10,54 | 0 | 13 |
| 10 | 0,07 | 0,57 | 0,04 | 2,61 | 0 | 3 |
| 11 | 0,08 | 0,70 | 0,08 | 1,73 | 0 | 0 |
| 12 | 0,00 | 0,00 | 0,00 | 0,14 | 0 | 0 |
| 13 | 0,01 | 0,60 | 0,73 | 1,05 | 0 | 0 |
| 14 | 0,08 | 0,97 | 0,02 | 0,52 | 1 | 1 |
| 15 | 0,00 | 0,28 | 0,05 | 3,30 | 0 | 2 |
| 16 | 0,27 | 0,64 | 0,06 | 1,70 | 0 | 0 |
| 17 | 0,11 | 0,15 | 0,02 | 0,50 | 0 | 0 |
| 18 | 0,00 | 0,00 | 0,00 | 0,00 | 0 | 0 |
| 19 | 0,00 | 0,05 | 0,00 | 0,11 | 0 | 0 |
| 20 | 0,00 | 0,01 | 0,00 | 0,02 | 0 | 0 |
| 21 | 0,02 | 0,09 | 0,01 | 0,35 | 0 | 0 |

890

**Table 6:** Time of concentration for the studied catchments: 5 methodologies have been used and the mean value has been chosen as representative in table 4.

| Catchment number | Pasini (m) | Ventura (m) | Pezzoli (m) | Kirpich (m) | NRCS-SCS (m) | Mean value (m) |
|---|---|---|---|---|---|---|
| 1 | 109.0 | 105.5 | 82.1 | 47.0 | 69.6 | 82.6 |
| 2 | 43.8 | 40.9 | 35.5 | 24.7 | 35.8 | 36.1 |
| 3 | 108.8 | 108.3 | 77.5 | 45.0 | 67.1 | 81.3 |
| 4 | 103.6 | 115.1 | 59.3 | 36.6 | 52.3 | 73.4 |
| 5 | 34.7 | 35.3 | 23.6 | 18.0 | 36.1 | 29.5 |
| 6 | 43.7 | 42.4 | 32.9 | 23.3 | 34.2 | 35.3 |
| 7 | 145.3 | 131.6 | 125.2 | 65.1 | 83.2 | 110.1 |
| 8 | 38.2 | 32.2 | 38.2 | 26.1 | 34.3 | 33.8 |
| 9 | 33.9 | 32.9 | 25.5 | 19.1 | 24.4 | 27.2 |
| 10 | 102.4 | 94.3 | 85.2 | 48.4 | 58.0 | 77.6 |
| 11 | 49.9 | 48.7 | 37.1 | 25.5 | 36.6 | 39.6 |
| 12 | 46.9 | 48.2 | 31.4 | 22.5 | 23.2 | 34.4 |
| 13 | 46.1 | 45.6 | 33.3 | 23.5 | 30.3 | 35.8 |
| 14 | 66.1 | 67.1 | 45.4 | 29.8 | 42.2 | 50.1 |
| 15 | 75.1 | 70.6 | 59.9 | 36.9 | 33.0 | 55.1 |
| 16 | 117.2 | 111.1 | 92.0 | 51.4 | 55.5 | 85.4 |
| 17 | 32.1 | 29.0 | 27.9 | 20.5 | 23.4 | 26.6 |
| 18 | 30.5 | 27.5 | 26.6 | 19.8 | 25.9 | 26.1 |
| 19 | 21.2 | 19.4 | 17.7 | 14.5 | 25.0 | 19.6 |
| 20 | 22.0 | 19.8 | 19.3 | 15.4 | 25.4 | 20.4 |
| 21 | 84.5 | 78.3 | 69.4 | 41.3 | 52.5 | 65.2 |

895

**Table 7:** The priority scales - A: using all the parameters; B: using parameters $k$, $l$ and $m$ (ref. Tab. 24); C: using parameters $a$ through $j$ (Tab. 2); D: using all the parameters and weighting the  elements to risk ones (tab 3).

| Catchment number | Priority scale A | Priority scale B | Priority scale C | Priority scale D |
|---|---|---|---|---|
| 1 | 5 | 5 | 4 | 4 |
| 2 | 4 | 4 | 4 | 3 |
| 3 | 2 | 3 | 1 | 1 |
| 4 | 4 | 4 | 3 | 3 |
| 5 | 4 | 5 | 3 | 3 |
| 6 | 4 | 4 | 3 | 3 |
| 7 | 5 | 5 | 4 | 4 |
| 8 | 2 | 2 | 2 | 1 |
| 9 | 1 | 1 | 2 | 2 |
| 10 | 4 | 4 | 4 | 4 |
| 11 | 5 | 4 | 4 | 4 |
| 12 | 5 | 4 | 5 | 5 |
| 13 | 3 | 3 | 2 | 3 |
| 14 | 4 | 4 | 4 | 4 |
| 15 | 4 | 3 | 4 | 4 |
| 16 | 4 | 4 | 4 | 3 |
| 17 | 5 | 4 | 5 | 4 |
| 18 | 5 | 5 | 5 | 5 |
| 19 | 5 | 5 | 4 | 5 |
| 20 | 5 | 5 | 5 | 5 |
| 21 | 5 | 4 | 5 | 5 |

| Priority scale |
|---|
| 1 |
| 2 |
| 3 |
| 4 |
| 5 |

900

**FIGURES**

[Figure]

905 Figure 1:

[Figure]

Figure 2:

910

[Figure]

Figure 3:

[Figure]

915

Figure 4:

[Figure]

Figure 5:

920

[Figure]

Figure 6:

[Figure]

925

[Figure]

Figure 7:

[Figure]

930    Figure 8:

[Figure]

Figure 9: